# Neuromorphic electro-stimulation based on atomically thin semiconductor for damage-free inflammation inhibition

Rong Bao[1,4], Shuiyuan Wang [2,4] ✉, Xiaoxian Liu[2,4], Kejun Tu [3], Jingquan Liu[3], Xiaohe Huang[2], Chunsen Liu [2], Peng Zhou [2] ✉ & Shen Liu[1] ✉

Inflammation, caused by accumulation of inflammatory cytokines from immunocytes, is prevalent in a variety of diseases. Electro-stimulation emerges as a promising candidate for inflammatory inhibition. Although electro-acupuncture is free from surgical injury, it faces the challenges of imprecise pathways/current spikes, and insufficiently defined mechanisms, while non-optimal pathway or spike would require high current amplitude, which makes electro-stimulation usually accompanied by damage and complications. Here, we propose a neuromorphic electro-stimulation based on atomically thin semiconductor floating-gate memory interdigital circuit. Direct stimulation is achieved by wrapping sympathetic chain with flexible electrodes and floating-gate memory are programmable to fire bionic spikes, thus minimizing nerve damage. A substantial decrease (73.5%) in inflammatory cytokine IL-6 occurred, which also enabled better efficacy than commercial stimulator at record-low currents with damage-free to sympathetic neurons. Additionally, using transgenic mice, the anti-inflammation effect is determined by β2 adrenergic signaling from myeloid cell lineage (monocytes/macrophages and granulocytes).

Inflammation, the response of the human body to noxious stimuli, is present in trauma, infection, surgery, burns, ischemia or necrotic tissue[1], which induces symptoms such as heat, redness, swelling, pain and loss of function[2]. Specifically, acute inflammation after tendon injury is a representative symptom caused by the accumulation of inflammatory cytokines from immunocytes[3]. The outcome refers to failure of tendon healing and ultimately peritendinous adhesions, limiting the patient's motor function and interfering with their daily life[4]. Reducing acute inflammatory cytokines can suppress peritendinous inflammation, but pharmacological treatment alone is inefficient[5,6] and has cardiovascular and genitourinary side effects[7]. Biomaterial-assisted loading therapy can enhance medicated efficacy

and mitigate side effects by controlling the release of carriers[8], but still has problems with tendon re-rupture, infection and severe foreign body inflammatory reactions[8].

Electro-stimulation (ES) of peripheral nerves is emerging as an anti-inflammatory therapy that intervenes in inflammatory cell activity by modulating bioelectrical signals, neurotransmitters to bring about relief from inflammation[9]. Vagal nerve stimulation initially became a common option[10–12], where afferent vagal fibers are stimulated to transmit signals and signals reach the target organ via efferent vagal nerve or sympathetic trunk to regulate inflammation[13,14]. Typical stimulation currents are 0.25−0.8 mA[15,16], high currents may lead to tissue damage and complications. Currently, electro-acupuncture or manual

[1]Shanghai Sixth People's Hospital Affiliated to Shanghai Jiao Tong University School of Medicine, Shanghai 200025, China. [2]Shanghai Key Lab for Future Computing Hardware and System, School of Microelectronics, Fudan University, Shanghai 200433, China. [3]National Key Laboratory of Science and Technology on Micro/Nano Fabrication, DCI Joint Team, Collaborative Innovation Center of IFSA, Department of Micro/Nano Electronics, Shanghai Jiao Tong university, Shanghai 200240, China. [4]These authors contributed equally: Rong Bao, Shuiyuan Wang, Xiaoxian Liu. ✉e-mail: sy_wang@fudan.edu.cn; pengzhou@fudan.edu.cn; liushensjtu@sjtu.edu.cn

acupuncture is commonly used for stimulation, and although it is minimally invasive, this non-directional stimulation requires higher current amplitudes. However, electro-acupuncture stimulation would not cause severe damage to neurons, as stimulation needles are usually placed in somatic tissue, away from the nerve bundle. While the lack of direct nerve stimulation reduces efficacy and leads to the demand for higher stimulation currents. Furthermore, as a mixed cranial nerve, vagal nerve consists of four-type nerves and is associated with a variety of functions that may induce dysfunction when stimulated, leading to complications including headache, nasopharyngitis, dyspnea, adverse cardiac and vocal fold paralysis[12,13]. The key to reducing complications is damage-free stimulation of non-mixed nerves.

As the efferent nerve from nervi spinales, sympathetic nerve shows a simple fiber composition with a convinced connection with anti-inflammation[17]. Attempts have been made at sympathetic ES, such as apical splenic nerve in the abdomen[18], which inhibited inflammation and reduced clinical symptoms in mice with rheumatoid arthritis, but there is still a blank in terms of tendon injury and its efficacy. Moreover, infection is a common complication of invasive stimulators and the risk is substantially increased by laparotomy[19]. Sympathetic chain ganglia (SChG) are paravertebral ganglia that mainly contain postganglionic fibers and act as efferent nerves innervating to the target organ. The lumbar segment of SChG runs between diaphragm and psoas major muscle and is accompanied by abdominal aorta in retroperitoneal space, which is a potential space behind the peritoneum. This pathway promises to reduce complications by direct sympathetic stimulation with less damage than laparotomy. Although non-invasive indirect stimulation of mouse sympathetic nerves by electroacupuncture through sensory nerves has no surgical damage[18,20], it faces challenges including imprecise pathways, uncontrolled efficacy, and insufficiently defined mechanisms, and stimulation current as high as 3 mA also cause nerve damage. Inappropriate stimulation paths would increase the risk of infection, as well as introducing high currents that inevitably cause nerve damage, whether invasive or non-invasive.

In addition, non-optimal stimulation spikes can further induce nerve damage and lead to complications. Complementary metal oxide semiconductor-based commercial stimulators provide stimulation pulse with fixed, abrupt amplitude[20,21]. These pulse waveforms are not programmable and differ from bioelectrical signals, the additional bionics and programmability introduced would increase area and power cost at the expense of implantability[22,23]. For stimulus efficacy, the low bionic pulse waveform usually needs to be compensated with a higher current amplitude, which is detrimental to the target nerve. Neuromorphic stimulation with bionic spikes promises therapeutic benefits at low currents and reduced biological damage[24], and has shown great success when acting as an afferent or efferent nerve for muscle contraction and limb movement[25,26]. Two-dimensional (2D) atomically thin semiconductors, represented by molybdenum disulfide ($MoS_2$), are flourishing in the fields of bionic sensor electronics[27–29] and neuromorphic engineering[30,31] due to their attractive properties. Excellent electrostatics within an ultra-thin thickness allows for low operating voltage and energy consumption[32]. More appealingly, 2D devices feature flexible programmability under external multi-physical fields and are capable of outputting bionic spikes for neuromorphic stimulation[33], as well as being biocompatible and exhibiting potential for implantability and scalability[34].

Here, we demonstrated an atomically thin $MoS_2$ floating-gate memory (FGM) interdigital circuit (IDC)-based neuromorphic ES that acts on sympathetic chain to inhibit inflammation in tendon injury. We isolated the sympathetic chain in retroperitoneal space and delivered direct stimulation to the segment of injured tendon by wrapped flexible electrodes. The constructed $MoS_2$ FGM shows excellent device performance and features biomimetic programmability, which mimics the progressively tunable long-term plasticity in biological synapses and fires bionic spikes to minimize nerve damage. The inflammation-

associated cytokines IL-6 was decreased by 73.5% in response to 2D ES. Benefiting from nerve-direct stimulation and bionic spikes, 2D ES exhibited better anti-inflammatory efficacy than a commercial stimulator, which used record-low currents with average amplitudes of ~0.175 mA to achieve a further 70.6% decline in IL-6 and damage-free to sympathetic neurons. Additionally, a gene knockout technique was adopted to verify that the β2 adrenergic receptor (ADRB2) on myeloid cell lineage (monocytes/macrophages and granulocytes) mediated the reduction of inflammatory cytokines under sympathetic ES. This work provides a direct, neuromorphic stimulation solution with minimal nerve damage, bringing new inspiration for inflammation inhibition and complication reduction.

## Results

### Neuromorphic Direct ES for Inflammation Inhibition in Tendon Injury

Figure 1 presents a schematic of the proposed neuromorphic direct ES to inhibit acute inflammation in injured tendons. 2D FGM IDC based on atomically thin monolayers of $MoS_2$ acts as a neuromorphic stimulator for firing bionic stimulus spikes (Fig. 1, bottom right). The 2D FGM IDC can be electrically connected to the flexible printed circuit (FPC) and implantable flexible electrodes by an anisotropic conductive film and inserted into the adapter PCB for connection to the analyzer/monitor (Fig. S1). Subsequently, the sympathetic nerve chain was isolated and wrapped with flexible stimulus ports, which enabled nerve-direct stimulation, as shown in the top panel of Fig. 1. Direct stimulation with bionic spikes promotes the release of noradrenaline (NE) from sympathetic nerve, which acts on the receptors on the myeloid cell lineage (monocytes/macrophages and granulocytes) surrounding the injured tendon, thus inhibiting cytokines associated with acute inflammation (Fig. 1, bottom left).

### Atomically Thin MoS₂-Based FGM IDC for Neuromorphic 2D ES

Control gate and the $Al_2O_3/HfO_2/Al_2O_3$ (A/H/A) stacked dielectric were sequentially deposited on a polyimide (PI) flexible substrate, followed by transferring monolayer $MoS_2$ films to form an IDC and spin-coating of polydimethylsiloxane (PDMS) encapsulation layer for neuromorphic ES (Fig. 2a). The detailed preparation flow is described in Supplementary Section 2. The prepared stimulator exhibits robust flexibility and can wrap around the target nerve (The magnified optical image is shown in Fig. S3a). The physical layout of 2D FGM IDC is shown in the lower panel of Fig. 2a, which includes 30 μm thick PI, control gate, the A/H/A dielectric stack, $MoS_2$ channels, interdigital electrodes, ground (GND) and stimulus ports. Figure 2b illustrates the device structure of monolayer $MoS_2$ FGM, Fig. 2c further shows a high-resolution cross-section scanning transmission electron microscopy (TEM) image (as indicated by the red dashed line in Fig. 2a) and the corresponding energy dispersive spectroscopy (EDS) mapping, which reveals a clear device interface distribution, including the monolayer $MoS_2$ channel and the A/H/A dielectric with thicknesses of 7/8/30 nm. Figure 2d displays a scanning electron microscopy (SEM) image of the central region in the 2D FGM IDC (The magnified SEM image is shown in Fig. S3b). The yellow dashed boxes are the interdigital electrodes (source/drain, connected to the stimulus port/adapter printed circuit board (PCB), respectively, for a total of 21 pairs), and the blue dashed boxes in the middle of the interdigital electrodes are the $MoS_2$ channels (width/length = 2). Control gates are present below each channel, as shown in the red dashed boxes. Raman spectrum of $MoS_2$ is shown in the top panel of Fig. 2e, which shows two peaks around 380, 408 cm$^{-1}$, corresponding to the $E^1_{2g}$ and $A_{1g}$ vibration modes (Raman spectra of the PI substrate are shown in Fig. S4a). In addition, atomic force microscopy (AFM) was used to characterize the channel morphology (Fig. S4b). The results showed that the monolayer $MoS_2$ channel had a uniform and flat surface with a thickness of ~0.7 nm, which laid the physical foundation for the excellent performance of

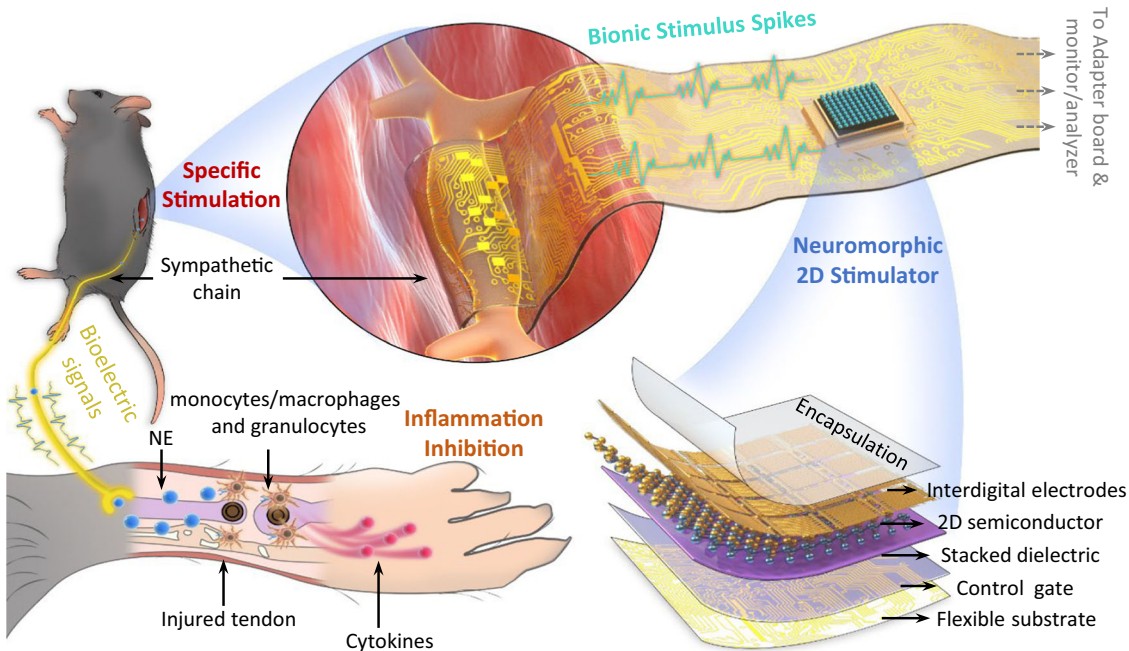

**Fig. 1 | Nerve-direct neuromorphic ES for inflammation inhibition in tendon injury.** Schematic of the proposed neuromorphic ES solution, where a flexible electrode is wrapped around sympathetic chain innervating the tendon segment to provide direct stimulation, as well as using 2D FGM IDC neuromorphic stimulator to fire bionic spikes. The combination of sympathetic-direct and bionic spikes promotes the release of NE, which inhibits the inflammatory cytokines associated with tendon injury while causing minimal nerve damage. NE: noradrenaline.

the 2D FGM IDC (Fig. 2e, bottom panel). We note that all the device components are fabricated in a scalable way, without the use of exfoliated materials. To evaluate the biocompatibility of the neuromorphic electrostimulator substrate (PI) and encapsulation (PDMS) materials when they are in contact with biological nerves and tissues, we cultured these materials with highly differentiated PC-12 cells (widely used in neuroscience, including studies on neurotoxicity, neuroprotection and neurosecretion) to imitate in-vivo conditions[35]. After coculture for 24 and 48 h, the proliferative capacity of the PC-12 cells in each group was tested by CCK-8 assay. As shown in Fig. 2f, the cell viability in both (PI, PDMS) coculture groups did not show significant differences compared to the control group (CTRL), which implies that PC-12 cells can maintain their proliferative capacity throughout the culture period. These results show the great biocompatibility and nonctytotoxicity of the implanted neuromorphic stimulator.

**Programmability and Biomimetic Spikes of 2D MoS$_2$ FGM IDC**

Figure 3 depicts the unit performance of a single channel 2D FGM in the neuromorphic ES IDC. The device transfer curves under different sweep ranges are shown in Fig. 3a, which exhibited a clear hysteresis window, with the hysteresis enables to widening as the sweep range increased. The gray curves represent 10 repetition sweeps from −15 V to +10 V, where the detailed transfer and leakage characteristics of the repeated scans are shown in Fig. S5a, b.The red curve is the mean curve, which shows good electrical robustness of the proposed 2D MoS$_2$ FGM. The speed characteristics are shown in Fig. 3b, with the device conductance (current) exhibiting reliable writing and erasure (±20 V) over the duration of 100 ns to 100 ms (pulses with varying widths are shown in Fig. S6). Before each operation, a pulse of +10 V was applied for 1 s to reset the device conductance to a high resistance state (HRS)[36]. Subsequently, the erase and write pulses were applied sequentially and the conductance underwent a transition from the initial HRS to a low resistance state (LRS) and back to a HRS. The 2D FGM modulates the conductance (switching between LRS and HRS) based on the mechanism of charge tunneling and capturing[37]. A

detailed energy band analysis can be found in Fig. S7. It is worth noting that with 100 ns write/erase pulse, the device still exhibited a non-volatile conductance difference (LRS/HRS) of nearly 2 orders of magnitude (Fig. 3b insert, see Fig. S8 for the detailed nonvolatile retention with varying pulse durations), indicating an ultrafast operation speed of 100 ns. Figure 3c demonstrates the endurance of MoS$_2$ FGM, which maintains significant conductance differentiation (LRS/HRS > 10) after 1000 cycles, despite slight degradation in durability with an increase in write/erase cycles (the original endurance curve is shown in Fig. S5c). The advantage of memory effect-based stimulation lies mainly in the greatly improved programmability, since 2D MoS$_2$ FGM IDC exhibits flexible biomimetic programmability. The application of successive erase and write pulses can cause progressively increasing and decreasing behaviors in device conductance (Fig. 3d), which mimics the long-term potentiation/depression (LTP/LTD) plasticity in biological synapses[31] (see Fig. S9 for the original LTP/LTD response curves under 30 gate pulses). The representative conductance multistates are shown in Fig. 3e, which suggest that the 2D FGM is programmed to perform nonvolatile stimulation over time. Another major advantage of memory effect is the realization of intrinsic neuromorphic bionic stimulation pulses, which effectively lower the stimulation current amplitude and biological damage. Specifically, MoS$_2$ FGM IDC stimulator is capable of generating bionic spikes, while commercial stimulators (such as Intan RHS2000) generally provide a fixed, abrupt biphasic rectangular current pulses (Fig. S10a). Although functional electrical stimulation may also employ sinusoidal or exponential waves for inducing sensation, muscle relaxation, etc., for nerve electrical stimulation, we focus on rectangular waves for comparison, considering the high efficiency of rectangular waves for activating nerve cells. Figure 3f presents the extracted intrinsic bioelectrical signals of sympathetic nerve (top panel, see Fig. S10b for detailed sympathetic nerve signals) and the bionic spikes (bottom panel) generated by a single-channel 2D FGM IDC under ultrafast gate pulses (see Fig. S11 for bionic spikes under different gate base voltages). Bionic spikes are similar to bioelectrical signals, and although not directly time-correlated, the frequency of bionic spikes can be flexibly regulated to match

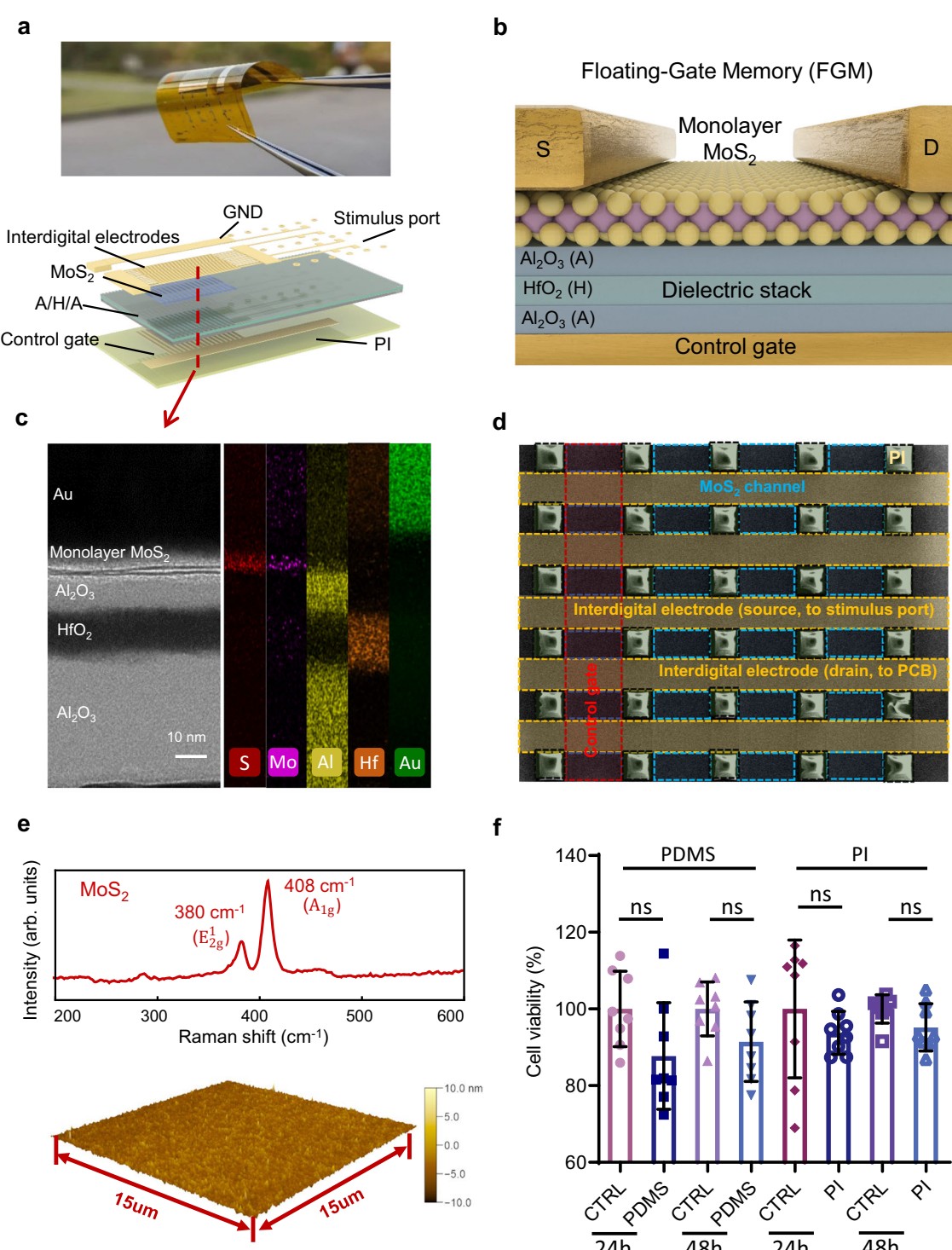

**Fig. 2 | Neuromorphic stimulator based on monolayer MoS₂ FGM IDC. a** Optical image and physical layout of the MoS₂ FGM IDC, which shows robust flexibility. GND: grounded, A/H/A: Al₂O₃/HfO₂/ Al₂O₃, PI: polyimide. **b**, **c** Device structure, cross-sectional TEM and the corresponding EDS element mapping of MoS₂ FGM. Scale bar: 10 nm. TEM was repeated three times with similar results. TEM: transmission electron microscope. **d** SEM image of the 2D FGM IDC, yellow, blue and red dashed boxes represent the interdigital (source/drain) electrode, MoS₂ channel and control gate, respectively. Scale bar: 50 μm. **e** Raman spectra of monolayer MoS₂, which shows two distinct peaks around 380, 408 cm⁻¹, corresponding to the $E^1_{2g}$ and

$A_{1g}$ vibration modes. AFM morphological characterization shows a uniform and flattened channel region with a thickness of ~0.7 nm. Scan window: 15 μm × 15 μm. **f** Biocompatibility tests. No significant different between the CTRL (control) and PDMS, PI groups after 24 and 48 h coculturing. Both PDMS and PI exhibited no cytotoxicity, implying good biocompatibility. $n = 8$ biologically independent experiments; two-sided Student's unpaired $t$-test. PDMS: polydimethylsiloxane group, CTRL: control group, PI: polyimide group, ns: no significance. Source data are provided as a Source Data file.

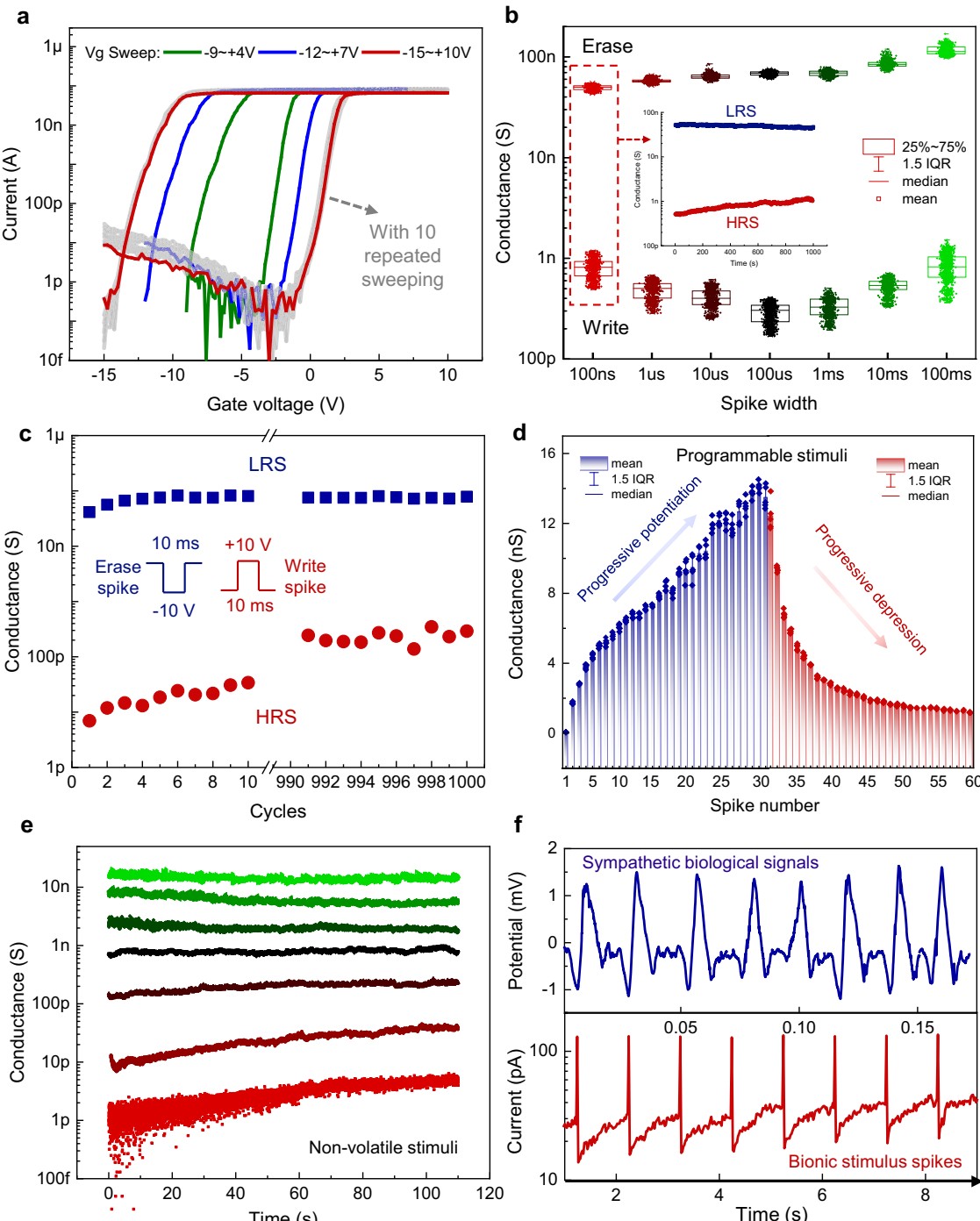

**Fig. 3 | Flexible programmability and bionic spikes of the 2D MoS₂ FGM. a** FGM transfer curves, which exhibit considerable hysteresis windows. The red line represents the mean curve of 10 repeated sweeps. Vg: gate voltage. **b** Boxplot of LRS/HRS for varying spike widths, which implies an ultrafast speed of 100 ns. The inset shows the nonvolatile retention (1000 s) at 100 ns. LRS: low resistance state, HRS: high resistance state, IQR: interquartile range. Box represents 25%−75% of conductance values, error bar represents 1.5 times IQR, horizontal line represents median, and hollow rectangle represents mean. **c** Endurance for 1000 cycles. The write/erase spikes are ±10 V, 10 ms. Squares and circles represent the mean

conductance of LRS and HRS at each cycle. **d** Device conductance shows a progressive increase and decrease, which mimics the LTP/LTD plasticity in biological synapses and enables programmable stimuli. Box represents mean, error bar represents 1.5 times IQR, and horizontal line represents median. **e** Representative conductance distribution after device programming (1 pA−10 nA). The programmed states exhibit nonvolatility. **f** Extracted sympathetic nerve signals (top panel) and bionic stimulation spikes (bottom panel) emitted by the MoS₂ FGM-based neuromorphic stimulator. Source data are provided as a Source Data file.

biological signals or targeted therapies, which provides potential to inhibit inflammation with lower stimulation amplitudes and minimal biological damage. Additionally, similar to existing stimulators, the bionic spikes generated by the FGM IDC allow for charge balancing, thereby minimizing tissue damage caused by charge accumulation and

electrolysis (Suplementary Section 12). And the neuromorphic electrostimulator consists of multi-channel MoS₂ FGM IDC. With the accumulation of interdigital channels, the increased current in the 2D FGM can meet the stimulation demand[38] (the climbing of the stimulation current is shown in Fig. S13a) and act as a neuromorphic

electrostimulator for inflammation inhibition. It is worth noting that this memory effect can be eliminated by removing the stacked A/H/A floating gate. With the assistance of peripheral FPC and adapter board, 2D FGM IDC can emit programmable bionic spikes for nerve stimulation in response to voltage pulses from the analyzer. The analyzer simultaneously displays the loop current of 2D FGM IDC in real time, i.e., real-time monitoring of nerve stimulation and providing feedback on the target nerve (see Fig. S13b, c for a typical real-time loop current at a stimulation frequency of 10 Hz and a duration of 15 min), while existing stimulators focus on settings that guarantee a fixed stimulus current amplitude and fail to monitor the current status in real time.

## Sympathetic ES on the Basis of SChG abdominal anatomy

The unique abdominal anatomy of SChG provides a convenient pathway for ES to regulate inflammation in tendon injuries. The sympathetic trunks consist of SChG that run on both sides of the thoracic spine and converge together after passing through the diaphragm in retroperitoneal space. Implanting electrodes through the retroperitoneal space could avoid all abdominal organs and reduce the risk of bleeding, which is safer and less invasive than other implantation methods. The anatomy of SChG has been verified by Heike et al. who performed whole-body histological staining and tissue removal in adult mice, allowing for remarkable visualization of the SChG and their physical connections[39]. Furthermore, the effector targets of the lumbar ganglia are generally the lower extremities[40,41]. We injected PRV-CAG-EGFP to the hind paw of wild type (WT, C57BL/6 J) mice and verified that ipsilateral L3 SChG innervated the hind paw (Fig. S14). We immunostained the sympathetic nerve on the section of the hind paw with an anti-TH (tyrosine hydroxylase) antibody, which showed the existence of sympathetic nerves around the flexor tendon (Fig. S15). This means that we can intervene on microenvironment of the hind paw by stimulating the lumbar SChG. Based on the above biological basis, we explored the regulatory effect of ES evoked by Intan RHS2000 commercial stimulator (namely traditional ES or t-ES, see Methods for details) on SChG to confirm that ES can inhibit inflammation after tendon injury. First, we supposed that the implantation of electrode had no significant influence to the hind paws. We chose IL-6 to characterize the changes in the microenvironment of hind paws since it is thought of as a typical cytokines involved in inflammation after tendon injury[42]. The details of electrode implantation are that we interfere with SChG in the lumbar segment through the retroperitoneal space (described in the Method and indicated by the yellow line in Fig. 4a, top panel). We isolated and wrapped it with an implantable flexible electrode for nerve-direct stimulation (Fig. 1 and Fig. S16). After implantation for 1 h, we simultaneously performed sham surgery (sham group, the detail of groups is in Methods) and tendon surgery (non-ES group) respectively on the left and right hind paws of WT (C57BL/6 J) mice (observing the influence of implantation on the same mouse). The tissues around the tendon were harvested for ELISAs after 0, 1, 2 and 3 h. The levels of IL-6 increased significantly at 2 and 3 h after tendon surgery, but remained low in the sham group at each timepoint (Fig. 4b). This suggests that implantation of electrode had little effect on inflammation and the inflammation around the tendon at these time points is primarily caused by tendon surgery. And the level of IL-6 in blood had no significant difference between the SHAM and sham groups (see detailed definitions in Methods), suggesting that the effect of implantation was negligible (Fig. S17). Next, the experimental strategy model was made according to the results above, including electrode implantation on the right lumbar SChG (Fig. S16), tendon surgery only on the right hind paws (excluding the influence of ES on the opposite hind paws), ES/non-ES and tissue collection for ELISA analysis and immunofluorescence (IF), as shown in Fig. 4a bottom panel. The duration of ES/non-ES was 15 min and the frequency was 10 Hz, as in previous studies[43]. Figure 4c showed that t-ES can reduce the level of IL-6 at 0.3 mA, which verified the feasibility of our ES model, whereas there is no inflammatory cytokine inhibition with 0.1 mA stimulation. After ES, lumbar SChG was harvested to perform

hematoxylin-eosin (HE) staining and TEM. Crevices were observed between neurons and connective tissues under the stimulation of 0.3 mA, while 0.1 mA caused less damage to sympathetic neurons (Fig. 4d). The swelling rough endoplasmic reticulum and low electron density of expended mitochondrion after 0.3 mA stimulation suggested that the higher stimulation current has caused damage to sympathetic neurons. In contrast, 0.1 mA is damage-free to neurons, similar to non-ES ones, which had been verified quantitatively by the maximum diameter and area of mitochondria (a strong indicator of cellular activity[44], Fig. 4e right panel, Fig. S18). Moreover, immunostaining for cleaved caspase-3 (CC3) on SChG showed that 0.3 mA caused apoptosis of SChG and in vitro experiments, CC3 staining and Calcein/PI test on PC12 cells were also proved that direct delivery at 0.3 mA caused damage to neurons (Fig. S19–21). The above results of HE and TEM showed that although 0.3 mA reduced inflammatory cytokines, it also introduced damage to SChG, which can lead to complications (It is worth pointing out that electro-acupuncture would not cause neuronal damage as severe as t-ES, since the stimulation needles are usually placed in somatic tissue and does not act directly on the nerve bundles). Lower currents or shorter stimulation durations are expected to tune down the damage caused by t-ES, but the efficacy may also be degraded. Therefore, we confirmed that sympathetic ES can inhibit inflammation following tendon injury, but this requires prevention of nerve damage.

## 2D ES for inhibiting inflammation in tendon injury with minimal nerve damage

In order to reduce nerve damage caused by high stimulation current, we performed ES using the 2D FGM IDC based neuromorphic stimulator (namely 2D ES, see Methods for details) and applied current stimulation with different amplitudes to find a suitable threshold range. ELISA statistics showed that currents with threshold of 0.04−0.6 mA (Fig. S22) reduced the levels of IL-6 by 73.5%, compared to the mean concentrations of them in non-ES group (Fig. 4c blue arrow). The IL-6 immunostaining was consistent with this result, as shown in Fig. 4f and Fig. S23a. The ELISA and immunostaining verified that 2D ES can slow down the process of inflammation. More appealingly, the neuromorphic 2D ES can achieve better efficacy than traditional commercial stimulators. As shown in Fig. 4c red arrow, the 2D ES group with average currents of -0.175 mA resulted in a further 70.6% drop in IL-6 levels compared to the t-ES group with fixed currents at 0.175 mA, which suggests that the direct neuromorphic bionic spikes lowered the threshold for ES efficacy (t-ES threshold is 0.3−2 mA). Moreover, the level of IL-6 had no significant statistical difference between t-ES of 0.3 mA and 2D ES of 0.175 mA, which means that 2D ES achieved the same effect of t-ES but with a 41.7% lower stimulation current. To gain a comprehensive analysis of inflammation after tendon injury, TNF-α, IL-10 and IL-1β were also tested[45–47]. The result showed that the level of TNF-α was not a good indicator because there is no significant difference between sham and non-ES group. But it was still decreased in the 2D ES group compared with non-ES group and the level of IL−10 and IL-1β were decreased in the 2D ES group as compared with the non-ES and 0.175 mA t-ES group, which fully analyzed the role of 2D ES in regulating the inflammatory-related cytokines (Fig. S24). Although acting as an anti-inflammatory cytokine, IL-10 also appeared to decrease after 2D ES, because in the previous study of suppressed inflammation, there is a significant reduction in inflammatory cytokine IL-6, and the elevation of anti-inflammatory cytokine IL-10 is limited. And IL-6 could promote the secretion of IL-10[48], the significantly reduced IL-6 may have led to the reduction in IL-10, which further suggests the effectiveness of 2D ES in inhibiting the progress of inflammation. In addition, NE is a neurotransmitter of sympathetic nerves. A significant increase in NE concentration of hind-paw tissue was observed in the 2D ES group compared to the sham and non-ES groups, suggesting that 2D ES activated the lumbar SChG and promoted NE secretion (Fig. 4g). Since NE was reported to decreasing IL-6 from splenic macrophages[49], then we

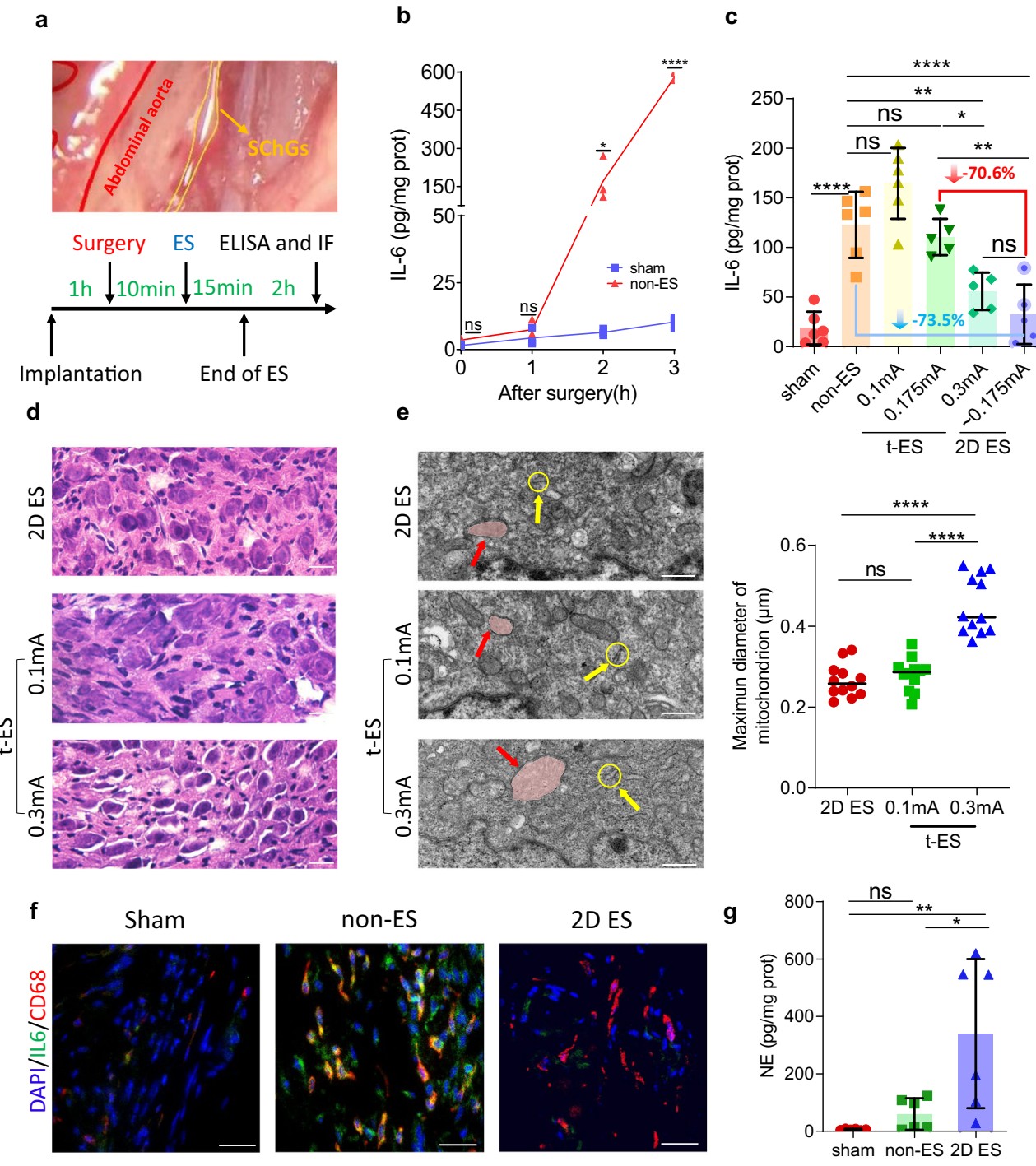

further investigated the cellular location of IL-6 in tendon injury. Immunostaining of IL-6 and CD68 showed that monocytes/macrophages infiltrated in the injured hind paw and IL-6 was secreted mainly from monocytes/macrophages. (Figs. 4f, S23d). The staining results also indicated that 2D ES prevented the release of IL-6 from monocytes/macrophages without any significant effect on the number of monocytes/macrophages, which was confirmed by the number of CD68[+] cells and the percentage of IL-6[+]CD68[+] cells among CD68[+]cells in the 2D ES and non-ES groups, as shown in Fig. S23b, c. Moreover, to rule out a potential anti-inflammation effect of 2D ES through blood flow of hind paws, laser speckle was performed and showed no significant changes in blood flow velocity before, during and after 2D ES with current in the threshold range (Fig. S25). The HE staining in Fig. 4d and the TEM in Fig. 4e and Fig. S18 further suggests that low stimulation currents of 2D

ES were expected to minimal nerve damage. Thanks to the bionic spikes as well as nerve-direct stimulation, the proposed MoS$_2$ FGM IDC-based neuromorphic 2D ES achieved greater efficacy of damage-free inflammatory inhibition (A performance benchmarks of the neuromorphic 2D ES versus existing stimulators are shown in Table S1, where bionic spikes refer to the range of spiking currents emitted by the multi-channel FGM IDC-based neuromorphic electrostimulator that are effective in reducing inflammatory cytokines).

## ADRB2 on myeloid cells triggers inflammatory inhibition in sympathetic ES

We further explored the biological mechanism of 2D ES in inhibiting inflammation and attempted to identify the mediating pathways. Since the rise of NE was observed after 2D ES, we tried to find out the

**Fig. 4 | 2D ES inhibits inflammation in tendon injury while minimal nerve damage. a** The anatomy of sympathetic trunk and ES strategy model. In the top panel, the red line is abdominal aorta and the yellow line is SChG. **b** The temporal variation of IL-6 in tissue around the injury tendon after surgery. ELISA test for IL-6 between sham and non-ES groups at 0, 1, 2, 3 h after surgery. It shows that IL-6 increased significantly at 2 h after surgery, which provided the timepoint of ES. $n = 3$ mice; two-sided Student's unpaired $t$ test; for 0 h, ns, $p = 0.0515$; for 1 h, ns, $p = 0.3224$; for 2 h, *$p < 0.05$; for 3 h, ****$p < 0.0001$. **c** ELISA for IL-6 in the non-ES, t-ES and 2D ES groups. The level of IL-6 was decreased by 73.5% in the 2D ES group compared with the non-ES group. The 2D ES group with average currents of -0.175 mA resulted in a further 70.6% drop in IL-6 level compared to the t-ES group with fixed currents at 0.175 mA, which suggests that the direct neuromorphic bionic spikes lowered the threshold for ES efficacy. And the level of IL-6 had no significant statistical difference between t-ES of 0.3 mA and 2D ES of 0.175 mA, which means that 2D ES achieved the same effect of t-ES but with a 41.7% lower stimulation current. n = 6 mice except for 0.175 mA t-ES and 2D ES groups which of n was 5 mice due to animal death; One-way ANOVA; $F_{5,27} = 25.89$, $p < 0.0001$; post hoc Tukey test: *$p < 0.05$; **$p < 0.01$; ****$p < 0.0001$; ns, $p = 0.1125$ (non-ES vs. 0.1 mA t-ES), $p = 0.973$ (non-ES vs. 0.175 mA t-ES), $p = 0.7459$ (2D ES). **d** The HE staining of lumbar SChG in the t-ES and 2D ES groups. 2D ES and 0.1 mA of t-ES causes little damage to sympathetic neurons, while 0.3 mA of t-ES shows crevices between neurons and connective tissues. Scale bar: 25 μm. $n = 3$ mice. **e** The TEM of sympathetic chain. The swelling rough endoplasmic reticulum (yellow arrow) and low electron density of mitochondrion (red arrow and colored shading) indicated that the 2D ES and 0.1 mA current has minimal damage to neurons compared to the 0.3 mA current. Maximum diameter of mitochondria after 0.3 mA stimulation was clearly larger than 2D ES and 0.1 mA, and the expansion of the mitochondria confirmed that a higher current caused damage to sympathetic neurons. $n = 12$; one-way ANOVA; $F_{2,33} = 45.99$, $p < 0.0001$; post hoc Tukey test: ****$p < 0.0001$; ns, $p = 0.7548$. Scale bar: 0.5 μm. **f** Immunostaining of DAPI, IL-6 and CD68. IL-6 and CD68 were co-expressed and the level of IL-6 declined after 2D ES. Scale bar: 100 μm. **g** ELISA for NE. NE was significantly increased after 2D ES. $n = 6$ mice; one-way ANOVA; $F_{2,15} = 8.189$, $p = 0.0039$; post hoc Tukey test: *$p < 0.05$, **$p < 0.01$, ns, $p = 0.8159$. NE: noradrenaline, ES: electro-stimulation, sham: sham surgery and flexible electrode implantation without ES group, t-ES: tendon surgery, flexible electrode implantation and traditional commercial ES based on the Intan RHS2000 commercial stimulator group, 2D ES: tendon surgery, flexible electrode implantation and neuromorphic ES based on 2D FGM IDC group, TEM: transmission electron microscope, HE: hematoxylin-eosin staining, ns: no significance. Source data are provided as a Source Data file.

category of adrenergic receptors in the hind paw of mice. The mRNA sequencing was tested in the peritendinous tissue from 2D ES and non-ES group, showing that the expression of the β2 adrenergic receptor (*Adrb2*) was significantly higher than *Adrb1*, *Adrb3*, α1 adrenergic receptor (*Adra1*) and *Adra2* both in the non-ES and 2D ES groups by screening out the differentially expressed genes (DEGs), as illustrated in Fig. 5a. Moreover, volcano plots were generated to identify the upregulated and downregulated genes based on the criteria of p value < 0.05 and |log (fold change) |>2. Figure 5b showed that the expression of *Adrb2* did not change between the non-ES and 2D ES groups, implying that the high expression of *Adrb2* was not caused by 2D ES, but may have been induced by tendon injury. The potential pathways were then investigated by Kyoto Encyclopedia of Genes and Genomes (KEGG) analysis for pathway enrichment, as shown in Fig. S26a. KEGG showed that the DEGs were mainly enriched in the cAMP signaling pathway in the 2D ES group compared with non-ES group, which is the main pathway of ADRB2[50]. Gene Ontology (GO) analysis revealed enrichment from three main aspects (Fig. S26b), including biological processes (BP), cellular components (CC) and molecular functions (MF). Regarding BP, the DEGs were mainly enriched in response to wounding, cellular divalent inorganic caution and cellular calcium ion homeostasis. In terms of CC, the DEGs mainly participated in cellular projections and integral components of the plasma membrane. Concerning MF, the DEGs were mainly involved in transporter activity and transmembrane transporter activity. Moreover, Simon et al. found that activation of the cholinergic anti-inflammatory pathway (CAP) by stimulation of vagus or splenic nerves in mice was mainly mediated by direct binding of NE to ADRB2 on splenic macrophages[51]. Therefore, we hypothesized that ADRB2 played an important role in inflammatory cytokine regulation via sympathetically-released NE. Since IL-6 was secreted mainly by monocytes/macrophages, immunostaining also showed the co-expression of ADRB2 and CD68 (Fig. 5d, Fig. S27). We further sought to clarify the inflammation related effect of ADRB2 on monocytes/macrophages after 2D ES. To selectively manipulate ADRB2-expressing monocytes/macrophages, we have developed the *Adrb2(flox/flox)* (*Adrb2(f/f)*) and *Lyz2-Cre* mouse lines (Fig. 5c). Figure 5d showed the ablation of ADRB2+ cells in monocytes/macrophages in the injured hind paw by immunostaining of CD68 and ADRB2. In *Lyz2-Cre::Adrb2(f/f)* mice, approximately 91% of the ADRB2+ monocytes/macrophages in the injured hind paw were ablated, compared to *Adrb2(f/f)* mice. The two types of mice were subsequently divided into 2D ES and non-ES groups according to whether neuromorphic 2D ES was performed after tendon injury. As showed in Fig. 5e, the level of IL-6 had no significant difference in two non-ES group. A decrease was

observed in *Adrb2(f/f)* mice after 2D ES, but 2D ES failed to decrease the level of IL-6 in *Lyz2-Cre::Adrb2(f/f)* mice, instead, it produced an increase in IL-6 compared with the non-ES group, which implies that 2D ES exerts its anti-inflammatory effects via ADRB2 on monocytes/macrophages. Immunostaining for IL-6 and CD68 validated the ELISA results, as displayed in Fig. 5f, g. We speculated that the increase in IL-6 was attributed to the presence of ADRA2, which may further promote inflammatory effects[52]. To verify the effect of ADRB2, we ablated the ADRB2 with zenidolol (ICI-118551) hydrochloride (a common antagonist against ADRB2) and repeated the experiment. Fig. S28 showed that the level of IL-6 was increased just as the *Lyz2-Cre::Adrb2(f/f)* mice with 2D ES. Similarly, Liu et al. electroacupunctured the acupoints after injection of LPS in mice and found ES promoted the inflammation. ES inhibited inflammation after ablation of ADRA2 with Yohimbine (a generic antagonist for ADRA2), which suggested that ES can play two opposing roles in regulating inflammation through ADRB2 and ADRA2[20]. Since ADRB2 was richer than ADRA2 in the hind paw of mice, so the effect of anti-inflammation dominated. After the ablation of ADRB2, the effect of pro-inflammation via ADRA2 dominated, which explained the increase of IL-6 in *Lyz2-Cre::Adrb2(f/f)* mice. Although we only tested the ablation of ADRB2 on monocytes/macrophages, there have been studies related to *Lyz2-Cre* mice that showed a deletion efficiency of 83−98% was determined in monocytes/macrophages and near 100% in granulocytes in the mice harboring both the *Lyz2-Cre* allele and *loxP*-flanked target genes[53]. Thus, it can be suggested that both monocytes/macrophages and granulocytes were involved in the inhibitory effect of ES on inflammation. In general, we essentially identified ADRB2 on myeloid cell lineage (monocytes/macrophages and granulocytes) as the trigger that mediates the reduction of inflammatory cytokine, although there appears to be a balance between the different receptors.

## Discussion

We have demonstrated a neuromorphic ES based on monolayer MoS$_2$ FGM IDC and wrapped sympathetic chain with flexible electrodes to perform direct stimulation for inhibiting acute inflammation that would lead to tendon adhesion. The 2D MoS$_2$ FGM showed remarkable performance, including ultrafast operation (100 ns), nonvolatile retention (1000 s) and robust durability (1000 cycles). MoS$_2$ FGM had biomimetic programmability that can mimic the progressive plasticity in biological synapses and fires neuromorphic bionic spikes. Subsequently, we isolated the sympathetic chain and delivered direct ES directly to the segment innervating the tendon. After implantation and artificial introduction of tendon injury, the inflammation-related

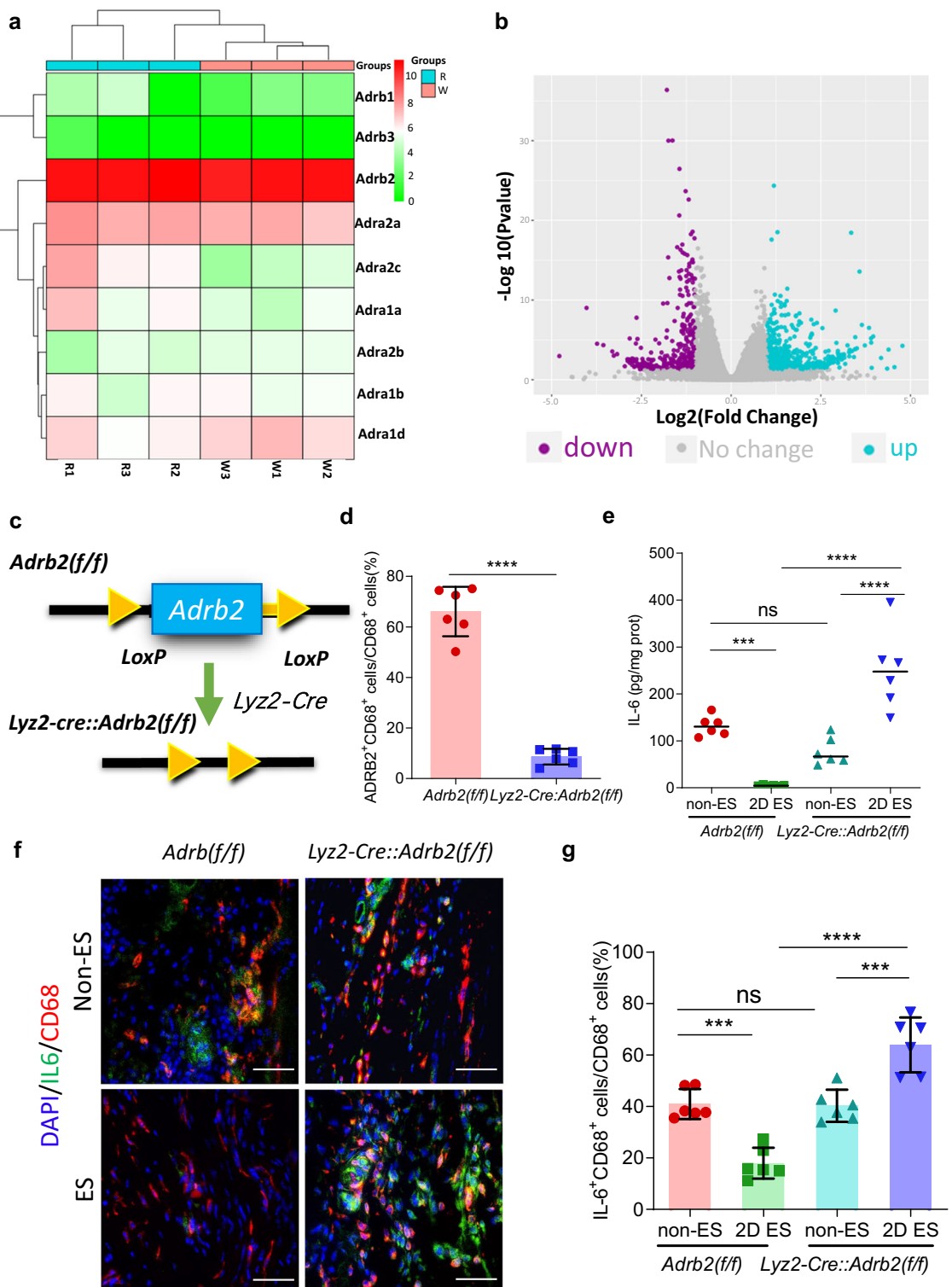

**Fig. 5 | ADRB2 on the myeloid cells triggers the anti-inflammatory efficacy in sympathetic ES. a** Heatmaps of RNA-sequencing results. R and W represent the 2D ES and non-ES groups, respectively. *Adrb2* was significantly enriched compared with the other adrenergic receptor genes. **b** Volcano plots of candidate DEGs in the microarray datasets based on the screening criteria. *Adrb2* showed no change between the non-ES and 2D ES groups. **c** Schematic of the intersectional genetic strategy used to generate *Lyz2-Cre::Adrb2(f/f)* mice. The statistical test was Wald test. **d** The percentage of ADRB2⁺CD68⁺ in the CD68⁺cells. These results verified the ablation of ADRB2 on monocytes/macrophages. *n* = 6 mice; two-sided Student's unpaired *t* test; ****$p < 0.0001$. **e** ELISA for IL-6 in the non-ES and 2D ES groups of *Lyz2-Cre::Adrb2(f/f)* and *Adrb2(f/f)* mice. The decrease of IL-6 was prevented after 2D

ES in *Lyz2-Cre::Adrb2(f/f)* mice. *n* = 6 mice; two-way ANOVA; $F_{1,20} = 63.53$, $p < 0.0001$; *p*ost hoc Tukey test: ***$p < 0.001$; ****$p < 0.0001$; ns, $p = 0.2134$. **f** Immunostaining of DAPI, IL-6 and CD68. Scale bar: 100 µm. **g** The percentage of IL-6⁺CD68⁺ cells among CD68⁺ cells. The results were in accordance with the ELISA data. *n* = 6 mice; two-way ANOVA; $F_{1,20} = 58.85$, $p < 0.0001$; post hoc Tukey test: ***$p < 0.001$; ****$p < 0.0001$; ns, $p = 0.9986$. ES: electro-stimulation, non-ES: tendon surgery and flexible electrode implantation without ES group, 2D ES: tendon surgery, flexible electrode implantation and neuromorphic ES based on 2D FGM IDC group, ns: no significance. *$p < 0.05$; **$p < 0.01$; ***$p < 0.001$; ****$p < 0.0001$. Source data are provided as a Source Data file.

cytokines IL-6 were reduced by 73.5% in mice with 2D ES compared to non-ES mice. In addition, thanks to the direct and programmable bionic stimulation spikes, the neuromorphic 2D ES exhibited better efficacy than a commercial stimulator (fixed at 0.175 mA), which produced a further 70.6% decrease in IL-6 after stimulation with the same duration (15 min) and damage-free currents (average amplitude of -0.175 mA). Additionally, we validated by gene knockout techniques that ADRB2 on myeloid cell lineage (monocytes/macrophages and granulocytes) mediated the decline of cytokines associated with inflammation in response to ES on sympathetic nerve. This work demonstrates the neuromorphic ES based on 2D FGM IDC and provides an effective strategy for treating inflammatory diseases while preventing complications caused by nerve damage. However, it is a fact that the proposed bionic stimulation solution is still in its infancy for practical clinical applications due to the limitations of scale integration in terms of devices and invasiveness in terms of biological surgery. And implantation of wirelessly powered flexible chips with integrated FGM circuits and electrodes via minimally invasive surgery, as well as non-invasive transcutaneous stimulation will be possible directions for neuromorphic bionic ES solution towards practical clinical applications.

## Methods

### Fabrication of MoS₂ FGM IDC for neuromorphic 2D ES

The PI (thickness of 30 μm) substrate was cleaned, followed by definition of the gate pattern using laser direct imaging (LDI) lithography (Durham Microwriter ML3), and deposition of the gate metal (Cr/Au, 5/25 nm) using electron beam evaporation (EBE, DETECH DE400). For the gate dielectric, atomic layer deposition (ALD, Beneq TFS200) was used to form A/H/A stacked layer at 150 °C with thicknesses of 7/8/30 nm. The chemical vapour deposited monolayer $MoS_2$ films (Six-Carbon Tech.) was transferred to the target locations on the PI substrates, and discrete channels by reactive ion etching (RIE, Sentch SI591). The interdigital electrode, contact sites for nerve stimulation and common ground patterns were defined by LDI lithography, and finally the Cr/Au (5/50 nm) metal was deposited again by EBE to form the neuromorphic electrostimulator circuit. PDMS was applied via spin coating for encapsulation insulation.

### Characterization and measurement of MoS₂ FGM IDC

Morphological characterization of the interdigital electrostimulator was carried out by optical microscopy (Olympus BX53M) and SEM (ZEISS GeminiSEM300). To examine the interface of the constructed 2D IDC, cross-sectional analysis was performed using TEM with EDS element mapping. In addition, the 2D channel of the electrostimulator was characterized by AFM (Oxford Instruments MFP-3D Origin+), showing a thickness of 0.7 nm for monolayer $MoS_2$. The monolayer $MoS_2$ channel and PI substrate were characterized by Raman spectroscopy, which showed strong peaks at 380, 408 cm⁻¹ and 1388, 1606, and 1785 cm⁻¹, respectively. A probe station (Cascade Summit11000B-M) equipped with a semiconductor analyzer (Keysight B1500A) was used to characterize the electrical properties of 2D FGM in the IDC. For in-vivo stimulation after implantation, the electrical connection between the IDC and FPC (20 Pins with 1 mm spacing) was realized by an anisotropic conductive film, and then connected to the Keysight B1500A analyzer via a customized PCB adapter for real-time monitoring of nerve stimulation.

### Experimental animals

*Lyz2-Cre* mice (B6.129P2-*Lyz2*^tm1(cre)lfo^/J, stock no: 004781) were purchased from The Jackson Laboratory. *Adrb2(fl/fl)* (B6/JGpt-Adrb2^em1Cflox^/Gpt, stock no: T052308) and C57BL/6 J (wild type, WT) mice were purchased from GemPharmatech. *Lyz2-Cre* mice were crossed with *Adrb2(fl/fl)* mice to obtain *Lyz2-Cre:: Adrb2(fl/fl)* mice. Mice of both sexes were utilized for the experiments unless specified. All the animals were housed in the specific pathogen-free (SPF) facility at the Animal Facility of Shanghai Sixth People's Hospital. The animals were exposed to a 12-h light/12-h dark cycle and provided food and water ad libitum. The ambient temperature was maintained between 20 °C and 26 °C, and the humidity was maintained between 40% and 70%.

### Tendon surgery with FGM IDC electrode implantation

The animal surgery strategy was described in Fig. 4a. More specifically, mice were anaesthetized by the administration of ketamine (60 mg/kg body weight) and xylazine (4 mg/kg body weight). The C57BL/6 J mice were divided into the SHAM (sham surgery only without electrode implantation), sham (sham surgery and flexible electrode implantation without 2D ES), 2D ES (tendon surgery, flexible electrode implantation and 2D ES), non-ES (tendon injury and flexible electrode implantation without ES) and t-ES (tendon surgery, flexible electrode implantation and traditional commercial ES based on the Intan RHS2000 commercial stimulator) groups. The *Adrb2(fl/fl) and Lyz2-Cre::Adrb2(fl/fl)* mice were divided into the 2D ES (tendon surgery, flexible electrode implantation and 2D ES) and non-ES (tendon surgery and flexible electrode implantation without 2D ES) groups. The mice were divided according to a random chart method. For tendon surgery, both sides of the flexor tendon were transected and repaired using Kessler sutures in the right hind-paw as in previous studies[54]. For electrode surgery, mice were fixed at a position of lateral decubitus. A 1-centimeter incision was made from the side of each mouse to further separate the muscles and tissues through retroperitoneal space. The sympathetic trunk running between the diaphragm and psoas major muscle was exposed behind the abdominal aorta. Then the right trunk was freed and flexible electrode (Microfab JETLAB 4, Microfab Technologies Inc., Shanghai, China) was wrapped in direct segment when ES was performed. The connectors at the end of the electrode could be connected to the external stimulator when needed. Finally, the incision was closed with a 6−0 suture. The animals were initially anaesthetized due to the need for surgery (ketamine, 60 mg/kg body weight and xylazine, 4 mg/kg body weight), but most animals were awake 1 h after electrical stimulation and no additional anesthesia was injected. The mice were humanely euthanized by intraperitoneal injection of pentobarbital (200 mg/kg) following the experiment.

### ELISA

The tissues around the tendon containing muscle and connective tissue were harvested for ELISAs. Every tissue was about 20 mg. 100 μl PBS was added to every tissue in the 1.6 ml EP tube and grinded using an ultrasonic grinder. Then the tissues were centrifuged at 5000 × g for 5 min at 4 °C and the supernatant was collected for ELISAs. The concentrations of IL-6 (MEC1008, Anogen), TNF-α (MEC1003, Anogen), IL-10 (A1010A0210, Biotnt) and IL-1β (MEC1010, Anogen) were analyzed by ELISA kits following the manufacturer's instructions. Briefly, 50 μL of standard or diluted sample are added to the appropriate well of the antibody pre-coated microtiter and then incubated for 1 h at room temperature. Plates are washed 5 times using an automatic plate washer. After final wash, invert plate, and blot dry by hitting plate onto absorbent paper or paper towels until no moisture appears. Add 50 μL anti-IL-6 biotin conjugate to each well and incubate for 1 h at room temperature. After washing, the avidin-HRP conjugate was added for 1 h of incubation at room temperature. After washing 5 times, the samples were incubated with 3,3",5,5"-tetramethylbenzidine (TMB) chromogenic reagent, in the dark for 15 min at room temperature. When the wells containing the standards exhibited a clear colored gradient, 100 μL of termination solution was added to stop the reaction. The absorbance of each well was measured at 450 nm. The associated protein concentration of each sample was calculated using the reference standard curve. NE (BA E-6200, LDN) was analyzed with an ELISA kit following the manufacturer's instructions similarly. Before analysis, noradrenaline (norepinephrine) was extracted using a cis-diol-specific affinity gel, acylated and then derivatized enzymatically.

## Injection of PRV and Zenidolol (ICI-118551) hydrochloride

The PRV-CAG-EGFP was purchased from BrainVTA (Wuhan) Co., Ltd. Briefly, 3ul PRV per mouse was injected to a tiny cut under the hind paw of WT. Six to seven days later, the mice were euthanized and the SChG were observed with fluorescent stereo microscope (Novel Optics, China). Zenidolol (ICI-118551) hydrochloride (2 mg/kg; 5 mg/ml in stock; HY-13951; MCE) was injected (i.p.) 30 min before electro-acupuncture stimulation.

## Immunofluorescent staining

Foot sections were first subjected to three 10−15-min rinses with 0.01 M PBS containing 0.3% Triton X-100 (A600198, BBI) (PBST). The samples were then blocked in the immunostaining blocking buffer (no. E674004, BBI) for 1 h. Subsequently, the samples were incubated with primary antibodies with in primary antibody dilution buffer (no. E674004, BBI) at 4 °C overnight. After three rinses with PBS, the samples were reacted with fluorescent dye-conjugated secondary antibodies in immunostaining secondary antibody dilution buffer (no. E674005, BBI) at RT for 1.5 h. The samples were then well rinsed three times and carefully mounted with DAPI Fluoromount-G (no. 0100-20, Southern Biotech). The primary antibodies used in this study included the following: rat anti-CD68 (NBP2-33337, Novus Biologicals; 1:400), rabbit anti-IL-6 (ab290735, Abcam; 1:50), rabbit anti-ADRB2 (ab182136, Abcam; 1:100), rabbit anti-Cleaved Caspase-3 (Cat. # 9661 S, CST; 1:400). Secondary antibodies conjugated to Alexa Fluor 488 (AF488), and AF594 included the following: AF488 donkey anti-rat (34406ES60, YEASEN) and AF594 donkey anti-rabbit (A21207, Invitrogen) and were diluted 1:400 unless otherwise specified.

## Hematoxylin-Eosin (HE) staining and TEM

The sympathetic chains slices were stained with HE Staining (Solarbio, Beijing, China). The sympathetic chain was scanned by the TEM (Joel JEM-1230), which were performed by Electron Microscope Technology Platform, Institue of Neuroscience, Chinese Academy of Sciences.

## Laser speckle imaging

After anesthesia, the hind paws were exposed and placed 30 cm below the laser speckle contrast imager (Simopto, Wuhan). Blood perfusion images were saved and analyzed by the Image Review software. A region of interest was selected in each experiment for measuring the blood flow velocity.

## Cell culture, Calcein/PI staining and CCK-8 assay for biocompatibility

Highly differentiated PC-12 cells (CSTR:19375.09.3101RATTCR9) were purchased from the National Collection of Authenticated Cell Cultures. The cells were cultured in high glucose DMEM supplemented with 10% foetal bovine serum, 100 μ/mL penicillin, and 100 μg/mL streptomycin at 37 °C in a humidified incubator with 5% $CO_2$. After three generations of proliferation, the progeny cells were used for experiments. Calcein/PI staining was supported by Calcein/PI Cell Viability/Cytotoxicity Assay Kit (C2015M, Beyotime) and the method is mentioned on section of immunofluorescent staining. Before coculture for CCK-8 assay, PI and PDMS were cut into 6-millimeter discs and placed into the bottom of a plate with 75% alcohol. The plate was placed under a UV lamp for 1 h. Then the discs were washed with PBS solution for 3 times and transferred to a 96-well plate (8 replicates were used for each material) at a density of $1 \times 10^5$ cells in each of the test groups (each well contained 100 μL of high glucose DMEM supplemented with 100 U/mL penicillin, and 100 μg/mL streptomycin). The control group consisted of cells without the above materials. After coculture for 24 and 48 h in the incubator, the materials and culture medium were removed and 100 μL of culture medium and 10 μL of CCK-8. Then the plate placed back into the incubator for 1 h. Finally, the viability of the cells was determined spectrophotometrically by measuring the absorbance at 450 nm.

## mRNA sequencing

The samples, namely the tissues around the tendon, were harvested from the hind paws of the animals 2 h after the end of electro-stimulation and were snap-frozen in liquid nitrogen for storage at −80 °C. Total RNA was extracted using Biozol Reagent (BW-R7311, BIOMIGA) according to the manufacturer's instructions. The concentration of RNA was determined using NanoDrop 2000 instrument (Thermo Scientific, USA). RNA quality inspection was carried out using Qubit 4.0 (Invitrogen, USA) and then 500 ng of 1% agarose electro-phoresis was used for detection. mRNA-seq was performed by Xu ran Biotech and included mRNA isolation and fragmentation, synthesis of double-stranded cDNA, PCR enrichment and quantification. The final sequencing was performed on the Illumina Novaseq 6000 (Illumina, USA).

## Statistical analysis

All statistical analyses were performed using Prism, software. Data were expressed as the mean ± standard deviation (SD). For pairwise comparisons (two samples involved), data were analyzed with two-sided Student's unpaired $t$ test. Data with multiple variables were analyzed using one-way ANOVA followed by post hoc Tukey's tests except for the analysis consisted with different ES modes on mice with different genotypes which were analyzed using two-way ANOVA followed by post hoc Tukey's tests. No statistical methods were used to predetermine sample sizes. Sample sizes for all histochemical, cytokine and transmitter measurements were chosen according to recently reported studies. Significant difference was considered as $p < 0.05$.

## Ethics statements

All mouse experiments were conducted in accordance with the guidelines of the Ethics Committee of Shanghai Sixth People's Hospital. The study was approved by the Ethics Committee of Shanghai Sixth People's Hospital (Reference Number: 2022-0688).

## Reporting summary

Further information on research design is available in the Nature Portfolio Reporting Summary linked to this article.

## Data availability

The data that support the findings of this study are available from the corresponding authors upon request. In addition, the RNA-seq data generated in this work have been uploaded in the GEO database under the accession code GSE253300. Source data are provided with this paper.

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

## Acknowledgements

This work was supported by the National Key Research and Development Program of China (2021YFA1200500 for P.Z.), National Natural Science Foundation of China (61925402, 62090032 for P.Z., 62304042 for S. Wang and No. 82172408, 81922045, 81772314 and 92368107 for S.L.), Science and Technology Commission of Shanghai Municipality (19JC1416600) (P.Z.), the Strategic Priority Research Program of the Chinese Academy of Sciences under grant XDB44000000 (P.Z.), China Postdoctoral Science Foundation (2022M720032) (S.W.), Shanghai Post-Doctoral Excellence Program (2022091) (S.W.) and Sailing Program (23YF1402100) (S.W.). The Original Exploration project (22ZR1480300) (S.L.), and the Outstanding Academic Leaders (Youth) project of Shanghai Science and Technology Innovation Action Plan (21XD1422900) (S.L.); Shanghai Municipal Health Commission (Grant No. 2022YQ073) (S.L.); Shanghai Jiao Tong University Medical College "Two-hundred Talent" Program (Grant No. 20191829) (S.L.). Shanghai Education Commission Shuguang plan (22SG09) (S.L.). Shanghai "Medical Star" Young Medical Talent Training Funding Program (Outstanding Youth) and Shanghai Health Commission Health Industry Research Project (Excellent Project) (S.L.).

## Author contributions

S.Wang and R.Bao co-designed and conducted the experiments with the assistance of X. Liu, K. Tu, J. Liu, X. Huang and C. Liu; S. Wang, S. Liu and P. Zhou conceived the idea and supervised the work; S. Wang and R. Bao co-wrote the manuscript. All authors provided suggestions for revisions and improvements to the work.

## Competing interests

The authors declare no competing interests.
