## [Peer Review File · Nature Communications]

REVIEWER COMMENTS

Reviewer #1 (Remarks to the Author):

In this study, Bao et al. developed a neuromorphic electro-stimulation (ES) method using an anatomically thin MoS₂ floating-gate memory interdigital circuit. They demonstrated that this neuromorphic ES can stimulate sympathetic neurons to control inflammation induced by tendon injury, at an intensity that does not cause nerve damage, unlike traditional ES that requires higher intensity and may cause some nerve damage. However, there are several issues to consider.

The main challenge is the invasive surgery required for delivering the device to surround sympathetic chain ganglia. This contrasts with electro-acupuncture and manual acupuncture, which are minimally invasive techniques.

Furthermore, while direct delivery of traditional ES to the sympathetic trunk ganglia might cause some nerve damage, the extent and nature of this damage were not thoroughly characterized. Additionally, most electro-acupuncture and manual acupuncture are performed in somatic tissue, with stimulation needles placed far away from the ganglia and not directly onto the nerve bundles. It remains unclear whether traditional ES techniques cause neuronal damage.

Regarding page 14, if 2D ES induced significant norepinephrine (NE) release that might cause vasoconstriction, it is puzzling why this release did not result in a change in blood flow.

In Figure 5g, to demonstrate that the enhanced interleukin-6 (IL-6) release by 2D ES is mediated via alpha-2 adrenergic receptor (ADRA2) in mice lacking the beta-2 adrenergic receptor (ADRB2), it would be necessary for the authors to test the effect of antagonists against ADRB2.

Reviewer #2 (Remarks to the Author):

This manuscript by Rong Bao et al. entitled 'Neuromorphic Specific Electro-Stimulation Based on Atomically Thin Semiconductor for Damage-Free Inflammation Inhibition', explores a fascinating strategy for neuromorphic electro-stimulation using a two-dimensional (2D) atomically thin semiconductor

floating-gate memory (FGM) interdigital circuit in mice. This innovative approach seeks to address inflammatory disorders through sophisticated electrostimulation techniques. The authors have centered their study on acute inflammation following tendon injury, a representative symptom arising from the accumulation of inflammatory cytokines, primarily from macrophages, among immune cells. They effectively demonstrate the capacity of this approach to significantly downregulate IL-6 secretion after tendon injury through the β -2 adrenergic receptor on myeloid cells.

Notably, this study presents a groundbreaking development by revealing that neuromorphic electrostimulation has the potential to inhibit inflammation in mice, expanding its application beyond its established role in restoring coordinated motions in mice with neurological motor disorders through bionic spikes.

While the physiological outcomes of this research align with some previously published studies, this study introduces novel elements. Specifically, the point of stimulation, focusing on the sympathetic chain, as well as the mechanism of action involving the β -2 adrenergic receptor on myeloid cells, presents a novel contribution to the field. These aspects have not been explored in prior literature.

In summary, this study holds significant promise and excitement for advancing the realm of bioelectronic medicine through the introduction of a unique technical tool for the electrostimulation-based treatment of inflammatory diseases.

To enhance the manuscript before potential publication, I have outlined several points of concern and suggestions:

Major Points:

1. From my point of view, one of the main weaknesses of the paper is the lack of comprehensive analysis of inflammatory cytokines, which is essential to validate the decrease in IL-6 attributed to electrostimulation. Including measurements of TNF, as the gold standard in the area of inhibition of inflammation by neuromodulation and a well-established marker in studies related to tendon injury (see as example PMID: 33451873, 31358843), and IL-10/IL-1 β , commonly associated with peripheral nerve electrostimulation (e.g. PMID: 29883598, PMID: 29883598), would strengthen the study's findings.
2. While there is a trend toward lower amount of IL-6 when looking at t-ES vs 2D ES, the claim that 2D ES demonstrates superior efficacy compared to a commercial stimulator requires stronger statistical substantiation (fig 4c, comparison between t-ES 0.3mA and 2D ES 175 μ A is lacking). This is an important point since the absence of significant would void any conclusions about the comparison of efficacy between t-ES and 2D ES.

3. The assertion of "specific electro-stimulation" by targeting the sympathetic chain needs further clarification, considering the presence of sensory nerves in this chain. Targeting the sympathetic chain is not sufficient to qualify the electrostimulation as 'specific' (table S1 needs to be corrected on this point see below). Unless the authors can show that the 2D ES stimulates specifically some subtypes of neural fibers inside the sympathetic chain ganglia.

4. The Methods section would benefit from more detailed explanations. For example:

a. "The tissues around the tendon were harvested for ELISAs", which tissue exactly (bones, muscle, adipose tissue...)? How much was collected? How were these processed (no information are available in the Method section about this point for IL-6 ELISA);

b. Animals were kept for 2 hours post-electrostimulation (figure 4a) but in which condition: anesthetized? Awake? If anesthetized how was the anesthesia maintained? In case of additional injection of Ketamine/Xylazine, which is highly anti-inflammatory: Which amount? Was the same amount of anesthetic injected across all animals? How is the amount of injected anesthetic controlled?

Minor Points:

5. The use of Lyz2-Cre mouse lines should be correctly associated with myeloid cells, not exclusively macrophages (abstract page 2 line 17 and in the main text).

6. Correct the plural form of "ganglion" to "ganglia" not "ganglias".

7. Page 4 line 4. 'Attempts have been made at sympathetic ES, such as apical spleen nerve in the abdomen (ref 18)'. Ref 18 is about pancreatic nerve not apical nerve. Gyuot et al. Brain Behav Immun. 2019 Aug;80:238-246 (PMID: 30885844) should be cited instead of ref 18.

8. Correct the typo "lumber segment" to "lumbar segment."

9. Ensure accurate citation of supplemental data in the order of appearance (e.g. S4 should be S1, S19 should be S16...)

10. Correct the reference in the text from "Fig 1b" to "Fig S1b."

11. Consider providing non-electrostimulated and non-implanted nerve controls for comprehensive assessment in figure 4.

12. Add a reference to "fig 4e" in the corresponding sentence (page 12, line 15).

13. Specify the applied ES profile (t-ES or 2D ES) in figure Page 14 line 2 and figure S18.

14. Provide more details about the tissue origin for IL-6 measurement in figure legends 4b and Methods section.

15. Table S1 need to be amended:

a. Microcuff electrodes were reported to be used by ref 5 for in vivo pancreatic nerve electrical stimulation not 'hook electrodes'.

b. In this table what does 'specificity' exactly means in that case? In my view, there is no apparent rationale for considering the microcuff electrodes utilized in reference 5 to be less specific than employing an electrode to encase the sympathetic chain, as demonstrated in the present study.

c. Excluding ultrasounds from this table is advisable, given the challenging nature of comparing electrical and ultrasound strategies in terms of amplitude, stimulus site, signal properties, and specificity.;

d. Please specify 'real-time monitoring of nerve stimulation' instead of 'real-time monitoring', which is confusing since it could refer to real-time monitoring of spontaneous sympathetic nerve activity.

16. Discuss the implications of the role of the β -2 adrenergic receptor on myeloid cells in light of Simon et al. 2023 (PMID: 37123375).

Reviewer #3 (Remarks to the Author):

The paper presents a neuromorphic electrical stimulator based on a two-dimensional (2D) ultra-thin semiconductor floating-gate memory (FGM) interdigital structure of an electronic circuit. The 2D floating-gate memory devices are presented to be programmable to mimic progressive synaptic plasticity and fire neuron-like spikes for stimulation, thus claiming to minimize nerve damage. The paper demonstrates the isolation of the sympathetic chain in retroperitoneal space and delivers customized stimulation to the segment of the injured tendon by using a flexible electrode array. This helps in decreasing the inflammatory response due to cytokine protein. The paper further claims to adopt a low stimulus magnitude with an average current of 175 μ A, which helps in minimizing damage to the sympathetic neurons. The authors also report that the proposed stimulation exerts anti-inflammation effects through macrophage β 2 adrenergic signaling by using mice subjects.

The paper is well-written and includes a wide range of measurements. I have personally liked the ideas of neuromorphic spike generation for stimulation, atomic-scale thin transistors with memory effects, and reduction of inflammation through low-magnitude stimulation. The diagrams are nice, and the paper seems thorough.

I have a few questions and concerns. The authors may consider addressing the following concerns:

How charge balancing is accomplished while adopting the presented neuromorphic stimulation pulses? I believe that charge balancing a must-have criterion for any electrical stimulation of the nervous system, which has not been addressed in the manuscript.

Can the authors show a detailed comparison between the threshold stimulus magnitude (or current) for both the bionic spike-like stimulus pulses and the standard biphasic pulses? What are the charge quantities per stimulus pulse while a specific gate bias voltage has been adopted?

Can the authors show the strength duration curve for the bionic stimulus waves? Can you state the measured chronaxie and rheobase?

How do you compare presented neuromorphic stimulus pulses with alternate stimulus patterns, such as exponential and other waves?

The reported stimulus pulses at various gate bias voltages in Figure S11 show a memory effect and a non-uniform pulse height over time. What are the specific advantages of memory effect-based stimulation? How can you remove such a memory effect in the future if such an effect is required to be ignored?

Electrode implantation in retroperitoneal space has been done for 1 hour only, which sounds inadequately short for assessing long-term effectiveness. Please justify the choice of this time duration and how it strongly correlates with long-term assessment.

The paper claims to have a low electrical stimulus current (175 μA average). However, the waveforms in Figure S11 show only nA currents.

Likewise, what is the correlation between Fig. 5f and Table S1? The current magnitudes are grossly different, although they represent the bionic spikes.

Optogenetic stimulus pulses are free from the charge delivery threshold and electrical stimulus artifacts? Can we consider such a non-electrical neural stimulation paradigm as a safer method than the reported stimulation scheme?

Other comments:

Fig S4 is a set-up photo and not a schematic.

Fig. S6: The captions are too small and unreadable. May be increased appropriately.

Fig. S7 seems to be manually drawn. Can you show a simulation result for the same?

Fig. S19: Scale bars are too small and almost invisible

Table S1: Why do you claim that real-time monitoring is incapable with other stimulators

Fig. 3f: The time axis is not labeled. Furthermore, is there a time correlation in between the blue and the red curves in this figure?

Response to Reviewers' Comments

Text in blue: Comments/questions from reviewers

Text in black: Response from the authors

Text in green: Text in the original submission

Text in red: Revisions made to the manuscript

Reviewer #1

General comments:

In this study, Bao et al. developed a neuromorphic electro-stimulation (ES) method using an anatomically thin MoS₂ floating-gate memory interdigital circuit. They demonstrated that this neuromorphic ES can stimulate sympathetic neurons to control inflammation induced by tendon injury, at an intensity that does not cause nerve damage, unlike traditional ES that requires higher intensity and may cause some nerve damage. However, there are several issues to consider.

Response:

We thank the reviewer for the valuable comments and suggestions on our work. As summarized by the reviewer, we proposed a neuromorphic ES based on MoS₂ floating-gate memory (FGM) interdigital circuit. Direct stimulation is achieved by wrapping the sympathetic nerve chain with flexible electrodes and the MoS₂ FGMs are programmable to mimic progressive synaptic plasticity and fire bionic spikes, which effectively reduces inflammation due to tendon injury with low nerve damage. Using transgenic mice, we further determined the biological mechanism by which ES exerts anti-inflammatory effects.

Regarding the main concern of invasive surgery required for implantation mentioned by the reviewer, we have **added a description of the used new pathway for less invasive implantation and a comparative discussion of the advantages over minimally invasive electro-acupuncture or manual acupuncture.** As for the extent

of damage caused by traditional ES (t-ES), we have **explained the definition of t-ES** and **add more in vitro and in vivo experiments to fully characterize neuron cell damage** to support our claim. In addition, we have **explained that elevated NE is not sufficient to cause vasoconstriction**, and **supplemented experiments with Zenidolol (ICI-118551) hydrochloride** (a common antagonist against ADRB2) to repeat the result in ADRB2-deficient mice. We have responded to the reviewers' comments point by point and made revisions in the main text and Supplementary Materials as suggested by the reviewers, which have been highlighted in yellow. **A summary of the revisions in response to the reviewers' comments can be found at the end of rebuttal letter.** We hope that the revised manuscript could address all your concerns.

For detailed comments, the main responses are summarized as follows:

- The new pathway used for less invasive device implantation was described and the advantages of our solution over electro-acupuncture were discussed, which had been added to the main text;
- The definition of t-ES was explained to avoid misunderstanding and more in vitro and in vivo experiments were added in **Supplementary Materials Section 19** to fully characterize the extent of damage;
- Explained the inability of elevated NE after 2D ES (compared to the non-ES group) to cause vasoconstriction (still below the threshold for vasoconstriction);
- Experiments on the effects of Zenidolol (ICI-118551) hydrochloride (a common antagonist against ADRB2) were added in **Supplementary Materials Section 26** to repeat the result in mice lacking ADRB2.

Comment 1:

The main challenge is the invasive surgery required for delivering the device to surround sympathetic chain ganglia. This contrasts with electro-acupuncture and manual acupuncture, which are minimally invasive techniques.

Response:

We thank the reviewer for the valuable comments. On the one hand, we **implanted the device through a potential space behind the peritoneum**, which is **much less invasive compared to existing implantation procedures**. On the other hand, although electro-acupuncture or manual acupuncture is minimally invasive, due to this **non-direct stimulation modality**, it **draws a much higher current amplitude** than direct stimulation of sympathetic chain ganglia, which could **lead to tissue damage**; meanwhile, electro-acupuncture or manual acupuncture **lacks direct stimulation on nerve**, which **lowers their effectiveness**. We have added further discussions of the new pathway used for less invasive implantation, as well as a comparison with electro-acupuncture (manual acupuncture) in the revised manuscript to highlight the advantages of our solution.

In contrast to existing invasive implantation procedures, we **used the sympathetic ganglion anatomy located in the retroperitoneal space for a less invasive device implantation**. More specifically, it is the posterior pararenal space (PPRS) formed by the Zuckerkandl fascia (i.e., posterior renal fascia, PRF) and transverse fascia (TF), as shown in **Figure R1a**. This pathway avoided all abdominal organs, **reducing the risk of bleeding, and the implanted mice survived for at least 4 weeks**. In addition, the results of blue line in **Figure 4b** showed that the **implantation had a negligible effect on the elevation of inflammatory cytokines** in hind paws. We had also **supplemented the level of IL-6 in blood between the SHAM (sham surgery only without electrode implantation) and sham (sham surgery and electrode implantation without ES) groups**, and the results were not significantly different (**Figure R1b**), further **suggesting that the effect of implantation was negligible**.

When it comes to electro-acupuncture (manual acupuncture is not discussed here, as electro-acupuncture has been proven to be superior to single manual acupuncture and is more widely accepted, **Ref 1**), even though it is non-invasive, **this non-directive modality of stimulation makes it necessary to have higher current amplitudes, which could also damage neurons**. In addition, **electro-acupuncture lacks direct**

stimulation on nerve compared to directly wrapped flexible electrodes, which **also reduces effectiveness and requires higher stimulus currents**. Liu et al. have demonstrated that a current of at least 3 mA is required to activate the sympathetic pathway using electro-acupuncture (Ref 2), which is **nearly 20 times higher than ours (~0.175 mA)**. Since we have confirmed that ganglia neuron cells will be injured under a stimulation current of 0.3 mA (commercial stimulator wrapped the nerve via flexible electrodes, **Figure 4d, e**), we can conclude that 3 mA stimulation with electro-acupuncture will **damage the cells and even burn the somatic tissue around the stimulation needles**. For that matter, the proposed neuromorphic stimulator requires lower current (~0.175 mA) and causes less damage by wrapping the nerve with flexible electrodes for direct stimulation.

In order to highlight the advantages of our proposed solution, we have further supplemented the discussions of the new pathway for less invasive implantation and a comparison with minimally invasive electro-acupuncture or manual acupuncture in the revised version.

Figure R1. a, Schematic of retroperitoneal space. The posterior pararenal space (PPRS) is consisted with Zuckerkandl fascia (i.e., posterior renal fascia, PRF) and transverse fascia (TF). LCF: lateral cone fascia. PRS: pararenal space. ARF: anterior

renal fascia. APRS: anterior pararenal space. PP: parietal peritoneum. **(Supplementary Figure 17) b, The level of IL-6 in blood before and after implantation.** The result between SHAM and sham groups showed that implantation had a negligible effect on the elevation of inflammatory cytokines. ns, no significance.

References:

1. Ulett GA, Han S, Han JS. Electroacupuncture: mechanisms and clinical application. *Biol Psychiatry*, 1998,44(2):129-38.
2. Liu S, Wang ZF, Su YS, Ray RS, Jing XH, Wang YQ, Ma Q. Somatotopic Organization and Intensity Dependence in Driving Distinct NPY-Expressing Sympathetic Pathways by Electroacupuncture. *Neuron*, 2020,108(3):436-450.e7.

Related discussions have been added to the revised manuscript:

(Page 11 line 17-22) **The unique abdominal anatomy of SChG provides a convenient pathway for ES to regulate inflammation in tendon injuries. The sympathetic trunk consists of SChG that run on both sides of the thoracic spine and converge together after passing through the diaphragm muscle present in retroperitoneal space. Implanting electrodes through the retroperitoneal space could avoid all abdominal organs, reduce the risk of bleeding, which is safer and less invasive than other implantation methods.**

(Page 3 line 18-Page 4 line 2) **Currently, electro-acupuncture or manual acupuncture is commonly used for stimulation, and although it is minimally invasive, this non-directional stimulation requires higher current amplitudes, which may also introduce damage. As electro-acupuncture is usually placed in the somatic tissue, away from the nerve bundle, there is a lack of direct stimulation of the nerves, which reduces effectiveness and requires higher stimulation currents. Typical stimulation currents for electro-acupuncture are 0.25-0.8 mA^{15,16}, high currents may lead to damage and cause complications.**

(Page 13 line 3-5) **And the level of IL-6 in blood had no significant difference between the SHAM and sham groups (see detailed definitions in Methods), suggesting that the effect of implantation was negligible (Fig. S17).**

(Page 21 line 18- Page 22 line 1, **Methods section**) **The C57BL/6J mice were divided into the SHAM (sham surgery only without electrode implantation), sham (sham surgery and flexible electrode implantation without 2D ES), 2D ES (tendon surgery, flexible electrode implantation and 2D ES), non-ES (tendon injury and flexible electrode implantation without ES) and t-ES (tendon surgery, flexible electrode implantation and traditional commercial ES based on the Intan RHS2000 commercial stimulator) groups.**

Comment 2:

Furthermore, while direct delivery of traditional ES to the sympathetic train ganglia might cause some nerve damage, the extent and nature of this damage were not thoroughly characterized. Additionally, most electro-acupuncture and manual acupuncture are performed in somatic tissue, with stimulation needles placed far away from the ganglia and not directly onto the nerve bundles. It remains unclear whether traditional ES techniques cause neuronal damage.

Response:

We thank the reviewer for the comments. **Traditional ES (t-ES) in our original manuscript refers to traditional commercial stimulator (such as Intan RHS2000) that delivers functional ES via flexible electrodes**, as compared to neuromorphic stimulation (i.e., 2D ES) based on 2D floating-gate memory interdigital circuits (FGM IDC), **does not refer to stimulation via electro-acupuncture or manual acupuncture** (electro-acupuncture versus direct nerve stimulation has been discussed in Comment 1). That is, t-ES also act directly on the ganglia, and higher stimulation currents can lead to neuronal damage. To avoid discrimination and misinterpretation,

we have **added a definition of “traditional ES (t-ES)” to distinguish it from electro-acupuncture or manual acupuncture stimulation.**

In the original manuscript, we have demonstrated t-ES damage to SChG cells by HE staining and TEM, and we agree with the reviewer that the SChG extent and nature of the damage should be thoroughly characterized and identified. We have **supplemented more in vitro and in vivo experiments to further characterize the damage of t-ES.** Cell apoptosis occurs with tissue injuries. Cleaved caspase-3 (CC3) staining is the common marker of cell apoptosis, and we have added immunostaining of these on SChG to assess the damage of t-ES (**Ref 1-2, Figure R2**). Since PC-12 cells are widely used in neuroscience, including studies on neurotoxicity, neuroprotection and neurosecretion (**Ref 3**). CC3 staining and Calcein/PI test were also supplemented in the in vitro experiments to assess the damage extent of PC-12 cells with t-ES (**Figure R3-5**). The result showed that **t-ES with stimulation current of 0.3 mA caused more cell apoptosis and more dead cells were observed**, compared to sham, t-ES at 0.1 mA and 2D ES. In addition, the damage for 2D ES on ganglia and PC-12 cells were also tested, the result showed that **2D ES caused little damage to PC-12 cells, similar to the sham and t-ES at 0.1 mA groups.**

(Supplementary Figure 19) Figure R2. The damage extent of neuron among the sham, t-ES (0.3, 0.1 mA) and 2D ES (~0.175 mA) groups. Cell apoptosis was observed after t-ES of 0.3 mA, while little apoptosis was observed in the other groups, suggesting that low currents are more suitable for biosafety. Scale bar: 50 μ m.

(Supplementary Figure 20) Figure R3. The damage extent of PC-12 among the sham, t-ES (0.3, 0.1 mA) and 2D ES (~0.175 mA) groups. CC3 staining of PC-12 cells. CC3 was expressed obviously in t-ES of 0.3 mA, while it was little in other groups, which means that low stimulation currents show better biosafety. Scale bar: 50 μm .

(Supplementary Figure 21) Figure R4. The damage extent of PC-12 among the sham, t-ES (0.3, 0.1 mA) and 2D ES (~0.175 mA) groups. Calcein/PI test of PC-12 cells. Calcein represents live cells and PI represents dead cells. PI⁺ cells was observed increasing in t-ES of 0.3 mA compared with the other groups, which also showed that higher current caused more damage to neurons. Scale bar: 50 μ m.

(Supplementary Figure 21) Figure R5. Statistics of PI+ cells / Calcein+ cells between the sham, t-ES (0.3, 0.1 mA) and 2D ES (~0.175 mA) groups. The percentage of PI⁺ cells/Calcein⁺ cells was observed increasing in t-ES of 0.3 mA compared with other groups, which indicated that higher current caused more damage to neurons. n=6. ns, no significance; ***, $p < 0.001$; ****, $p < 0.0001$.

Refernece:

1. Cho EA, Kim EJ, Kwak SJ, Juhn YS. cAMP signaling inhibits radiation-induced ATM phosphorylation leading to the augmentation of apoptosis in human lung cancer cells. *Mol Cancer*, 2014, 13:36.
2. D'Amelio M, Cavallucci V, Cecconi F. Neuronal caspase-3 signaling: not only cell death. *Cell Death Differ*, 2010, 17(7):1104-14.
3. Chen, P. et al. Wirelessly Powered Electrical-Stimulation Based on Biodegradable 3D Piezoelectric Scaffolds Promotes the Spinal Cord Injury Repair. *ACS nano*, 2022, 16, 16513-16528.

Related discussions have been added to the revised manuscript:

(Page 13 line 20- Page 14 line 1) Moreover, immunostaining for cleaved caspase-3 (CC3) on SChG showed that 0.3 mA caused apoptosis of SChG and in vitro experiments, CC3 staining and Calcein/PI test on PC12 cells were also proved that direct delivery at 0.3 mA caused damage to neurons (Figure S19-21).

(Page 24 line 12-15, **Methods section**) The primary antibodies used in this study included the following: rat anti-CD68 (NBP2-33337, Novus Biologicals; 1:400), rabbit anti-ADRB2 (ab182136, Abcam; 1:100), rabbit anti-Cleaved Caspase-3 (Cat. # 9661S, CST; 1:400).

(Page 25 line 7-15, **Methods section**) **Cell culture, Calcein/PI staining and CCK-8 assay for biocompatibility.** Highly differentiated PC-12 cells (CSTR:19375.09.3101RATTCR9) were purchased from the National Collection of Authenticated Cell Cultures. Calcein/PI staining was supported by Calcein/PI Cell Viability/Cytotoxicity Assay Kit (C2015M, Beyotime) and the method is mentioned on section of immunofluorescent staining. Before coculture for CCK-8 assay, PI and PDMS were cut into 6-millimeter discs and placed into the bottom of a plate with 75% alcohol.

**Related discussions have been added in Supplementary Materials Section 19:
Section 19. Comprehensive assessment of damage after ES.**

We have demonstrated t-ES damage to SChG cells by HE staining and TEM. To thoroughly characterize and identify extent and nature of the damage on the SChG after ES. We have supplemented more in vitro and in vivo experiments to further characterize the damage of t-ES (**Figure S19-21**). Cell apoptosis occurs with tissue injuries. Cleaved caspase-3 (CC3) staining is the common marker of cell apoptosis, and we have added immunostaining of these on SChG to assess the damage of t-ES. Since PC-12 cells are widely used in neuroscience, including studies on neurotoxicity, neuroprotection and neurosecretion. CC3 staining and Calcein/PI test were also supplemented in the in vitro experiments to assess the damage extent of PC-12 cells with t-ES.

Comment 3:

Regarding page 14, if 2D ES induced significant norepinephrine (NE) release that might cause vasoconstriction, it is puzzling why this release did not result in a change in blood flow.

Response:

We thank the reviewer for raising this important question. It has been reported that elevated NE does cause vasoconstriction (**Ref 1**), as mentioned by the reviewer. In this work, **the increase in NE in the 2D ES group is compared to the non-ES group**. However, the increased NE caused by 2D ES relative to the non-ES group did not lead to changes in blood flow, as shown by the laser speckle. We inferred that **the absolute value of increased NE failed to reach the threshold for vasoconstriction**. In other words, the 2D ES-induced increase in NE (~300 pg/mg prot) was **lower than** the level of NE required for vasoconstriction (**Ref 1**).

Reference:

1. Hadoke PW, Christy C, Kotelevtsev YV, et al. Endothelial cell dysfunction in mice after transgenic knockout of type 2, but not type 1, 11beta-hydroxysteroid dehydrogenase. *Circulation*. 2001;104(23):2832-2837.

Comment 4:

In Figure 5g, to demonstrate that the enhanced interleukin-6 (IL-6) release by 2D ES is mediated via alpha-2 adrenergic receptor (ADRA2) in mice lacking the beta-2 adrenergic receptor (ADRB2), it would be necessary for the authors to test the effect of antagonists against ADRB2.

Response:

Thank the reviewer for the valuable comments and suggestions. We have **supplemented the experiments on the effect of Zenidolol (ICI-118551) hydrochloride** (a common antagonist against ADRB2) **in Supplementary Materials Section 26** to repeat the result in mice lacking ADRB2.

Zenidolol hydrochloride was injected (i.p.) 30 min before 2D ES, with doses referred to previous reports (Ref 1). When ADRB2 was ablated by Zenidolol (ICI-118551) hydrochloride, we found that the level of IL-6 failed to decrease after 2D ES compared with non-ES group (Figure R6). Instead, IL-6 level was elevated, as shown by the results in *Lyz2-Cre::Adrb2(f/f)* mice (Figure 5e).

In summary, these results explained the increase of IL-6 in *Lyz2-Cre::Adrb2(f/f)* mice after 2D ES: norepinephrine inhibits inflammation through ADRB2; 2D ES inhibited inflammation because the high expression of ADRB2 surround the injured tendon shown by RNA-seq.

(Supplementary Figure 28) Figure R6. The level change of IL-6 under the intervene of adrenoceptor. ELISA for IL-6 in the non-ES and 2D ES groups of *Lyz2-Cre::Adrb2(f/f)* and *Adrb2(f/f)* mice. The decrease of IL-6 was prevented after 2D ES in *Lyz2-Cre::Adrb2(f/f)* mice. 2D ES also failed to decreased the level of IL-6 after the ablation of ADRB2 through Zenidolol hydrochloride. Instead, it was increased compared with non-ES group. n=5-6. ns, no significant; *, $p < 0.05$; ***, $p < 0.001$; ****, $P < 0.0001$.

Reference:

1. Archer T, Fredriksson A. Effects of clonidine and alpha-adrenoceptor antagonists on motor activity in DSP4-treated mice I: dose-, time- and parameter-dependency. *Neurotox Res*, 2000, 1(4):235-247.2.

Related discussions have been added to the revised manuscript:

(Page 18 line 2-5) To verify the effect of ADRB2, we ablated the ADRB2 with zenidolol hydrochloride (a common antagonist against ADRB2) and repeated the experiment. **Fig. S28** showed that the level of IL-6 was increased just as the *Lyz2-Cre::Adrb2(f/f)* mice with 2D ES.

(Page 23 line 18- Page 24 line 2, **Methods section**) **Injection of PRV and Zenidolol (ICI-118551) hydrochloride.** The PRV-CAG-EGFP was purchased from BrainVTA (Wuhan) Co., Ltd. Briefly, 3ul PRV per mouse was injected to a tiny cut under the hind paw of WT. Six to seven days later, the mice were euthanized and the SChG were observed with fluorescent stereo microscope (Novel Optics, China). Zenidolol (ICI-118551) hydrochloride (2 mg/kg; 5 mg/ml in stock; HY-13951; MCE) was injected (i.p.) 30 min before electroacupuncture stimulation.

Reviewer #2

General comments:

This manuscript by Rong Bao et al. entitled ‘Neuromorphic Specific Electro-Stimulation Based on Atomically Thin Semiconductor for Damage-Free Inflammation Inhibition’, explores a fascinating strategy for neuromorphic electro-stimulation using a two-dimensional (2D) atomically thin semiconductor floating-gate memory (FGM) interdigital circuit in mice. This innovative approach seeks to address inflammatory disorders through sophisticated electrostimulation techniques.....They effectively demonstrate the capacity of this approach to significantly downregulate IL-6 secretion after tendon injury through the β -2 adrenergic receptor on myeloid cells.

Notably, this study presents a groundbreaking development by revealing that neuromorphic electro-stimulation has the potential to inhibit inflammation in mice, expanding its application beyond its established role in restoring coordinated motions in mice with neurological motor disorders through bionic spikes.

While the physiological outcomes of this research align with some previously published studies, this study introduces novel elements. Specifically, the point of stimulation, focusing on the sympathetic chain, as well as the mechanism of action involving the β -2 adrenergic receptor on myeloid cells, presents a novel contribution to the field. These aspects have not been explored in prior literature.

In summary, this study holds significant promise and excitement for advancing the realm of bioelectronic medicine through the introduction of a unique technical tool for the electrostimulation-based treatment of inflammatory diseases.

To enhance the manuscript before potential publication, I have outlined several points of concern and suggestions.

Response:

We thank the reviewer for the careful reading and **highly positive comments on our work** (such as “This innovative approach seeks to address inflammatory disorders through sophisticated electrostimulation techniques”, “this study presents a groundbreaking development by revealing that neuromorphic electro-stimulation has

the potential to inhibit inflammation in mice”, “this study introduces novel elements.....presents a novel contribution to the field”, “this study holds significant promise and excitement for advancing the realm of bioelectronic medicine through the introduction of a unique technical tool for the electrostimulation-based treatment of inflammatory diseases”) as well as the **suggestions for potential publication in *Nature Communications***. As summarized by the reviewer, we proposed a neuromorphic ES strategy based on MoS₂ floating-gate memory (FGM) interdigital circuit (IDC). Direct stimulation was achieved by wrapping the sympathetic nerve chain with flexible electrodes, which was combined with MoS₂ FGMs programmable synaptic plasticity to fire bionic spikes, thereby downregulating inflammatory cytokines with minimal nerve damage. We have subsequently used transgenic mice to further define the biological mechanisms by which ES exerts anti-inflammatory effects.

Here, we have responded to the reviewer’s concerns and suggestions point by point: adding more comprehensive indicators of inflammatory cytokines, supplementing statistical comparisons (t-ES vs. 2D ES), replacing “specific stimulation” with “direct stimulation”, complementing the **Methods** section, correcting plural forms and typos, updating references, renumbering of supplementary data, revising **Table S1**, and adding relevant discussions (highlighted in yellow) in the revised main text and Supplementary Materials. **A summary of the revisions in response to the reviewers’ comments can be found at the end of the rebuttal letter.** We hope that the revised manuscript could address all your concerns.

For detailed comments, the main responses are summarized as follows:

- Supplemented experiments and added the results of TNF- α , IL-10 and IL-1 β in **Supplementary Materials Section 22** to enable a comprehensive analysis of inflammatory cytokines;
- Added statistical comparisons of t-ES at 0.3 mA, 0.175 mA with 2D ES at ~0.175 mA to the revised **Figure 4c** and added related discussions;

- Replaced the “specific stimulation” with “direct stimulation” to more accurately describe the stimulation mode, and revised the Title and **Table S1**;
- Added more detailed explanations in the **Methods** section, including tissues for ELISA, animal condition after ES;
- Replaced the “macrophages” with “myeloid cell lineage (monocytes/macrophages and granulocytes)”, and added a discussion related to the ablation validation of ADRB2;
- Corrected the plural form of “ganglion” to “ganglia” and “Sympathetic chain ganglia (SChGs)” to “SChG”, corrected the typo of “lumber segment” to “lumbar segment”;
- Replaced the original **Ref 18** with the work recommended by the reviewer and added relevant discussions;
- Renumbered the supplementary data according to the order of their appearance in the main text, and updated the **Supplementary Materials**;
- Corrected the reference to **Fig. 1b** in the original version to **Fig. 2a**;
- Supplemented non-implanted and non-electrostimulated group (named SHAM group), and compared with the sham group (with implanted electrodes but no electrical stimulation), while **no significant difference was found for IL-6 in peritendinous tissues**;
- Added reference for **Fig. 4e**, and additionally supplemented mitochondrial area as an indicator of neuron damage in **Supplementary Materials Section 18**;
- Revised **Figure S25** and the description in the manuscript to specify 2D ES used;
- Added the details of tissue origin in the caption of **Figure 4b** and **Methods** section.
- Revised **Table S1** as recommended, including correcting Ref 5 for electrode type (microcuff), revising the stimulation mode (replacing specificity with direct nerve stimulation), removing the ultrasound technique, and supplementing real-time monitoring of nerve stimulation;
- Added reference recommended by the reviewer and supplemented related discussions of β -2 adrenergic receptor on myeloid cells.

*****Major Points*****

Comment 1:

From my point of view, one of the main weakness of the paper is the lack of comprehensive analysis of inflammatory cytokines, which is essential to validate the decrease in IL-6 attributed to electrostimulation. Including measurements of TNF, as the gold standard in the area of inhibition of inflammation by neuromodulation and a well-established marker in studies related to tendon injury (see as example PMID: 33451873, 31358843), and IL-10/IL-1 β , commonly associated with peripheral nerve electrostimulation (e.g. PMID: 29883598, PMID: 29883598), would strengthen the study's findings.

Response:

We thank the reviewer for the valuable suggestions. We have **supplemented the experiments and added the results of TNF- α , IL-10 and IL-1 β in Supplementary Materials Section 22** to enable a comprehensive analysis of inflammatory cytokines.

We agree with the reviewer that more comprehensive inflammatory cytokines analyses are needed to validate the reduction of IL-6 by electrostimulation. In accordance with the reviewer's suggestion, we have performed supplementary experiments to test typical inflammatory markers including TNF- α , IL-10, and IL-1 β (**Ref 1-3**) between the sham group, non-ES group, t-ES and 2D ES groups with stimulation current both approximated at 0.175 mA. The results showed that a decrease in TNF- α , IL-10 and IL-1 β levels occurred in the 2D ES group compared to the non-ES group and the t-ES group, as shown in **Figure R7**. Supplementary experiments further **confirmed that the reduction in IL-6 was an effect of electrostimulation and also demonstrated the better inflammatory inhibition of 2D ES compared to t-ES.**

(Supplementary Figure 24) Figure R7. The level of TNF- α , IL-10, IL-1 β between sham, non-ES, t-ES (0.175 mA) and 2D ES (\sim 0.175 mA) groups. a, The level of TNF- α had no significance between sham and non-ES groups, while 2D ES still decreased it compared with the non-ES group. b-c, 2D ES declined the level of IL-10, IL-1 β compared with non-ES and 0.175 mA t-ES. n=5; *, $p < 0.05$; *, $P < 0.001$; ****, $P < 0.0001$. ns, no significance.**

Reference:

1. Hanaka M, Iba K, Hayakawa H, et al. Delayed tendon healing after injury in tetranectin-deficient mice[J]. Journal of Orthopaedic Science, 2022, 27(1): 257-265.
2. Best K T, Lee F K, Knapp E, et al. Deletion of NFKB1 enhances canonical NF- κ B signaling and increases macrophage and myofibroblast content during tendon healing[J]. Scientific reports, 2019, 9(1): 10926.
3. Komegae E N, Farmer D G S, Brooks V L, et al. Vagal afferent activation suppresses systemic inflammation via the splanchnic anti-inflammatory pathway[J]. Brain, behavior, and immunity, 2018, 73: 441-449.

The corresponding revisions in the revised manuscript:

(Page 14 line 22-Page 15 line 3) To gain a comprehensive analysis of inflammatory cytokines, TNF- α , IL-10 and IL-1 β were also tested. The result showed that the level of TNF- α , IL-10 and IL-1 β were decreased in the 2D ES group as compared to the non-ES and 0.175 mA t-ES, which fully analyzed the role of 2D ES in regulating the inflammatory cytokines (Fig. S24).

(Page 22 line 22-Page 23 line 2, **Method section**) The concentrations of IL-6 (MEC1008, Anogen), TNF- α (MEC1003, Anogen), IL-10 (A1010A0210, Biotnt) and IL-1 β (MEC1010, Anogen) were analyzed by ELISA kits following the manufacturer's instructions.

Comment 2:

While there is a trend toward lower amount of IL-6 when looking at t-ES vs 2D ES, the claim that 2D ES demonstrates superior efficacy compared to a commercial stimulator requires stronger statistical substantiation (fig 4c, comparison between t-ES 0.3 mA and 2D ES 175 μ A is lacking). This is an important point since the absence of significant would void any conclusions about the comparison of efficacy between t-ES and 2D ES.

Response:

We thank the reviewer for this valuable comment, and we agree with the reviewer that statistical confirmation is important for conclusions. We have **supplemented statistical comparisons of t-ES at 0.3 mA, 0.175 mA with 2D ES at ~0.175 mA to the revised Figure 4c and added related discussions.**

In the original **Figure 4c**, we compared t-ES and 2D ES groups statistically with the non-ES group, missing the statistical analysis between t-ES and 2D ES. And we agree with the reviewer that **statistical substantiation of t-ES and 2D ES is essential to the conclusions, especially for t-ES at 0.3 mA and 2D ES at ~0.175 mA.** As suggested by the reviewer, we have added a statistical comparison of t-ES at 0.3 mA with 2D ES at ~0.175 mA. The results showed **no statistically significant difference between t-ES at 0.3 mA and 2D ES at 0.175 mA (Figure R8),** which means that **2D ES achieved a similar effect as t-ES, but with a 41.8% lower stimulus current.**

In addition, to further directly show the better effect of 2D ES, we have also supplemented the t-ES group with the same stimulation current of 0.175 mA. The results showed a statistically significant difference between 2D ES and t-ES both at

0.175 mA (**Figure R8**), suggesting that 2D ES produces better anti-inflammatory efficacy than t-ES at the same stimulation current (i.e., 2D ES enables efficacy with lower currents).

(Revised Figure 4c) Figure R8. ELISA for IL-6 in the non-ES, t-ES and 2D ES groups. The level of IL-6 was decreased by 73.5% in the 2D ES group compared with the non-ES group. the 2D ES group with average currents of ~0.175 mA resulted in a further 70.6% drop in IL-6 levels compared to the t-ES group with fixed currents at 0.175 mA, which suggests that the specific neuromorphic bionic spikes lowered the threshold for ES efficacy. Moreover, the level of IL-6 had no significant statistical difference between t-ES of 0.3 mA and 2D ES of 0.175 mA, which means that 2D ES achieved the same effect of t-ES but with a 41.8% lower stimulation current. n=5-6. ns, no significance; *, $p < 0.05$; **, $p < 0.01$; ****, $p < 0.0001$.

The corresponding revisions in the revised manuscript:

(Page 14 line 16-22) **As shown in Fig. 4c red arrow**, the 2D ES group with average currents of ~0.175 mA resulted in a further 70.6% drop in IL-6 levels compared to the t-ES group with fixed currents at 0.175 mA, which suggests that the direct neuromorphic bionic spikes lowered the threshold for ES efficacy. Moreover, the level

of IL-6 had no significant statistical difference between t-ES of 0.3 mA and 2D ES of 0.175 mA, which means that 2D ES achieved the same effect of t-ES but with a 41.8% lower stimulation current.

(Page 34 line 6-12, **Figure caption of Figure 4c**) **c**, ELISA for IL-6 in the non-ES, t-ES and 2D ES groups. The level of IL-6 was decreased by 73.5% in the 2D ES group compared with the non-ES group. The 2D ES group with average currents of ~0.175 mA resulted in a further 70.6% drop in IL-6 level compared to the t-ES group with fixed currents at 0.175 mA, which suggests that the direct neuromorphic bionic spikes lowered the threshold for ES efficacy. And the level of IL-6 had no significant statistical difference between t-ES of 0.3 mA and 2D ES of 0.175 mA, which means that 2D ES achieved the same effect of t-ES but with a 41.8% lower stimulation current. n=5-6.

Comment 3:

The assertion of “specific electro-stimulation” by targeting the sympathetic chain needs further clarification, considering the presence of sensory nerves in this chain. Targeting the sympathetic chain is not sufficient to qualify the electrostimulation as ‘specific’ (table S1 needs to be corrected on this point see below). Unless the authors can show that the 2D ES stimulates specifically some subtypes of neural fibers inside the sympathetic chain ganglia.

Response:

We thank the reviewer for this valuable reminder. In the revised version, we have followed the reviewer’s suggestion to **replace the “specific stimulation” with “direct stimulation”** to more accurately describe the stimulation mode, and updated the Title and Table S1.

“Specific stimulation” in the original manuscript meant that the ES was performed directly on the nerve. However, although the proportion of sensory nerve fibres in the sympathetic chain is small (the vast majority of which are sympathetic,

Ref 1), we still cannot rule out the potential influence of sensory nerves. Therefore, as the reviewer mentions, the description of “specific stimulation” may be inappropriate. To more accurately describe the mode of electro-stimulation, we **replaced “specific” with “direct” to emphasize that the flexible electrode wraps around the sympathetic chain for direct electro-stimulation** (distinct from the indirect stimulation of electro-acupuncture or manual acupuncture, which performs in somatic tissue, with stimulation needles placed far away from the ganglia and **not directly onto the nerve bundles**).

Reference:

1. Wehrwein EA, Orer HS, Barman SM. Overview of the Anatomy, Physiology, and Pharmacology of the Autonomic Nervous System. *Compr Physiol.* 2016;6(3):1239-1278.

The corresponding revisions in the revised version:

We have revised all relevant descriptions of “specific stimulation” to “direct stimulation” in the main text and Supplementary Materials, also including Title, Table S1, etc.

(Page 1 line 1-2) **Revised Title: Neuromorphic Electro-Stimulation Based on Atomically Thin Semiconductor for Damage-Free Inflammation Inhibition.**

Revised Table S1 (Revisions are highlighted):

Electrostimulator system	Microcuff electrode	Electroacupuncture	Electroacupuncture	Commercial stimulators	2D Neuromorphic Electrostimulator
Targeted symptom (diseases)	Autoimmune diabetes	Systemic inflammation	Systemic inflammation	Inflammation in tendon injuries	Inflammation in tendon injuries
Stimulus site	Near pancreatic nerve	Acupoints (ST36, ST25)	Acupoints (ST36, ST25)	Sympathetic chain	Sympathetic chain
Direct nerve stimulation	Yes	No	No	Yes	Yes
Stimulus signal	Fixed rectangular charged-balanced biphasic pulses	Fixed rectangular charged-balanced biphasic pulses	Fixed rectangular charged-balanced biphasic pulses	Fixed rectangular charged-balanced biphasic pulses	Programmable bionic spikes
Current amplitude	0.45 mA	0.5-3 mA	0.5 mA	0.3-3 mA	average of ~0.175 mA (0.04-0.6 mA)
Real-time monitoring of nerve stimulation	incapable	incapable	incapable	incapable	capable
References	Nat. Biotechnol. 2019 [5]	Neuron 2020 [6]	Nature 2021 [7]	Our work	Our work

Comment 4:

The Methods section would benefit from more detailed explanations. For example:

- a. “The tissues around the tendon were harvested for ELISAs”, which tissue exactly (bones, muscle, adipose tissue...)? How much was collected? How were these processed (no information is available in the Method section about this point for IL-6 ELISA);
- b. Animals were kept for 2 hours post-electrostimulation (figure 4a) but in which condition: anesthetized? Awake? If anesthetized how was the anesthesia maintained? In case of additional injection of Ketamine/Xylazine, which is highly anti-inflammatory: Which amount? Was the same amount of anesthetic injected across all animals? How is the amount of injected anesthetic controlled?

Response:

We thank the reviewer for the kind suggestions. In the revised version, we have **added more detailed explanations in the Methods section**. Here we divided the reviewer’s questions into two parts (in blue) and answered them individually (in black).

a. “The tissues around the tendon were harvested for ELISAs”, which tissue exactly (bones, muscle, adipose tissue...)? How much was collected? How were these processed (no information is available in the Method section about this point for IL-6 ELISA)”

The tissue contained muscle and connective tissue and are approximately 20 mg per tissue. 100 µl of PBS was added to every tissue in a 1.6 ml EP tube and grinded using an ultrasonic grinder. The tissues were then centrifuged at 5000× g for 5 minutes at 4°C and the supernatant was collected for use in ELISA. The above description has been added in the revised manuscript.

The corresponding revisions in the revised manuscript:

(Page 22 line 19-22, **Methods section**) **ELISA.** The tissues around the tendon containing muscle and connective tissue were harvested for ELISAs. Every tissue was about 20 mg. 100 µl PBS was added to every tissue in the 1.6 ml EP tube and grinded using an ultrasonic grinder. Then the tissues were centrifuged at 5000× g for 5 minutes at 4°C and the supernatant was collected for ELISAs.

b. “Animals were kept for 2 hours post-electrostimulation (figure 4a) but in which condition: anesthetized? Awake? If anesthetized how was the anesthesia maintained? In case of additional injection of Ketamine/Xylazine, which is highly anti-inflammatory: Which amount? Was the same amount of anesthetic injected across all animals? How is the amount of injected anesthetic controlled?”

As required for surgery, all groups of animals were first anesthetized with ketamine (60 mg/kg body weight) and xylazine (4 mg/kg body weight). That is, **the animals were initially anaesthetized, but most of them were awake 1 hour after electro-stimulation and no further anesthesia was injected.** Apart from the surgery, no additional anesthesia was injected throughout and **the procedure was the same for all animals.**

In addition, **Figure 4b** shows a gradual increase in IL-6 levels at 0, 1, 2, and 3 hours after tendon surgery, which suggests that **anesthesia did not have a significant inhibition effect on inflammatory cytokines.** We have also added a statement on animal anesthesia in the revised manuscript.

The corresponding revisions in the revised manuscript:

(Page 22 line 13-16, **Methods section**) **The animals were initially anaesthetized due to the need for surgery (ketamine, 60 mg/kg body weight and xylazine, 4 mg/kg body weight), but most animals were awake 1 hour after electrical stimulation and no additional anesthesia was injected.**

*****Minor Points*****

Comment 5:

The use of *Lyz2-Cre* mouse lines should be correctly associated with myeloid cells, not exclusively macrophages (abstract page 2 line 17 and in the main text).

Response:

We thank the reviewer for this kind reminder. When referring to *Lyz2-Cre*, we have replaced the “macrophages” with “myeloid cell lineage (monocytes/macrophages and granulocytes)”, and supplemented a discussion related to the ablation validation of ADRB2.

Lyz2-Cre mouse is typically used for Cre-lox studies of the myeloid cell lineage (monocytes/macrophages and granulocytes). As mentioned by reviewer, it should be correctly associated with myeloid cell lineage, not only with macrophages, which we have revised. And in this work, co-immunostaining of CD68 and ADRB2 was performed to verify the ablation in *Lyz2-Cre::Adrb2(f/f)* mice. **CD68 is the marker of monocytes/macrophages but not the marker of granulocytes**, which means it may fail to prove the ablation of ADRB2 in myeloid cell lineage. However, it has been reported that a deletion efficiency of 83-98% was determined in monocytes/macrophages and near 100% in granulocytes in the mice harboring both the *Lyz2-Cre* allele and *loxP*-flanked target genes tested (Ref 1). Therefore, **CD68 could indicate the ablation of ADRB2 in myeloid cell lineage containing monocytes/macrophages and granulocytes**. A relevant description of ablation validation has been added to the revised manuscript.

Reference:

1. Clausen B E, Burkhardt C, Reith W, et al. Conditional gene targeting in macrophages and granulocytes using *LysMcre* mice[J]. Transgenic research, 1999, 8: 265-277.

The corresponding revisions in the revised manuscript:

(Page 18 line 12-20) Although we only tested the ablation of ADRB2 on monocytes/macrophages, there have been studies related to *Lyz2-Cre* mice that showed a deletion efficiency of 83-98% was determined in monocytes/macrophages and near 100% in granulocytes in the mice harboring both the *Lyz2-Cre* allele and *loxP*-flanked target genes. Thus, it can be suggested that both monocytes/macrophages and granulocytes were involved in the inhibitory effect of ES on inflammation. In general, we essentially identified ADRB2 on myeloid cell lineage (monocytes/macrophages and granulocytes) as the trigger that mediates the reduction of inflammatory cytokine, although there appears to be a balance between the different receptors.

Comment 6:

Correct the plural form of “ganglion” to “ganglia” not “ganglias”.

Response:

We thank the reviewer for the careful reading and for correcting our spelling errors. In the revised manuscript, we have corrected the plural form of “ganglion” to “ganglia” and “Sympathetic chain ganglias (SChGs)” to “SChG”.

Comment 7:

Page 4 line 4. ‘Attempts have been made at sympathetic ES, such as apical spleen nerve in the abdomen (ref 18)’. Ref 18 is about pancreatic nerve not apical nerve. Gyuot et al. *Brain Behav Immun.* 2019 Aug; 80:238-246 (PMID: 30885844) should be cited instead of ref 18.

Response:

We thank the reviewer for this kind reminder. In the revision, we have replaced the original Ref 18 with the work recommended by the reviewer and added relevant discussions.

Specifically, Guyot et al. found that electrical stimulation of the apical nerve enabled suppression of inflammation and mitigated clinical symptoms in mice with rheumatoid arthritis (**Ref 1**). They further demonstrated that the inhibition of inflammation by apical nerve electrical stimulation relied on signalling by both β 2-ARs and α 7nAchRs in myeloid cells, with these two signalling pathways acting in parallel (**Ref 1**). To accurately describe apical nerve electrical stimulation, we have replaced the original **Ref 18** with this work.

References:

1. Guyot M, Simon T, Panzolini C, et al. Apical splenic nerve electrical stimulation discloses an anti-inflammatory pathway relying on adrenergic and nicotinic receptors in myeloid cells[J]. Brain, behavior, and immunity, 2019, 80: 238-246.
(Newly replaced reference to the original Ref 18)

The corresponding discussions have been added in the revised manuscript:

(Page 4 line 8-11) **Attempts have been made at sympathetic ES, such as apical splenic nerve in the abdomen¹⁸, which inhibits inflammation and reduces clinical symptoms in mice with rheumatoid arthritis, but there is still a blank in terms of tendon injury and its efficacy.**

Comment 8:

Correct the typo “lumber segment” to “lumbar segment”.

Response:

Thanks to the reviewer for reminding us to correct typos. In the revised manuscript, **we have corrected the original “lumber segment” to “lumbar segment”**. And the full manuscript has been double-checked.

Comment 9:

Ensure accurate citation of supplemental data in the order of appearance (e.g. S4 should be S1, S19 should be S16...).

Response:

We thank the reviewer for this kind suggestion. In the revised version, we have **renumbered the supplementary data according to the order of their appearance in the main text, and updated the Supplementary Materials.**

Comment 10:

Correct the reference in the text from “Fig 1b” to “Fig S1b”.

Response:

We thank the reviewer for this kind reminder. **The reference to Fig 1b in the original version should be to Fig. 2a, which we have corrected.**

The original **Fig. 1b** refers to the cross-section of scanning transmission electron microscopy and the corresponding energy dispersive spectroscopy mapping in **Fig. 2c**. After several figure revisions, we forgot to update the figure reference, which led to this error. Actually, **Fig. 1b** here should be **Fig. 2a**, and we have corrected it. We again thank the reviewer for your careful reading and kind reminders.

The corresponding revisions in the revised manuscript:

(Page 7 line 17-21) **Fig. 2c** further shows a high-resolution cross-section scanning transmission electron microscopy (TEM) image (as indicated by the red dashed line in **Fig. 2a**) and the corresponding energy dispersive spectroscopy (EDS) mapping, which reveals a clear device interface distribution, including the monolayer MoS₂ channel and the A/H/A dielectric with thicknesses of 7/8/30 nm.

Comment 11:

Consider providing non-electrostimulated and non-implanted nerve controls for comprehensive assessment in figure 4.

Response:

We thank the reviewer for this suggestion. In addition to the sham group (implanted electrodes but no electrical stimulation), we have **supplemented non-implanted and non-electrostimulated group (named SHAM group), which was compared with the sham group, and found that there was no significant difference of IL-6 levels in peritendinous tissues.**

In the original manuscript, we used the implanted electrodes but without electrical stimulation as the control group, i.e., the sham group. **But it doesn't seem to rule out the potential effect of electrode implantation.** As suggested by the reviewer, we further supplemented the sham group with non-electrostimulated and non-implanted group (that is, only sham surgery, named the SHAM group) to more comprehensively analyse the effects of electrical stimulation. Experimental results showed that **the level of IL-6 in peritendinous tissues of the SHAM group were not significantly different from those of the sham group (Figure R9),** suggesting that electrode implantation had no effect on peritendinous tissues.

In other words, for **Figure 4**, there is no need to add redundant SHAM group as other control group, however, we have **supplemented a discussion of the SHAM group definition and its negligible effect on implantation (through IL-6 levels in blood, Figure S17)** in the revised version.

Figure R9. IL-6 level in peritendinous tissues of the SHAM and sham groups. The level of IL-6 in peritendinous tissues in the SHAM group had no significant difference with the sham group, which means electrode implantation had no effect on peritendinous tissues. ns, no significance.

The corresponding revisions in the revised manuscript:

(Page 13 line 3-5) **And the level of IL-6 in blood had no significant difference between the SHAM and sham groups (see detailed definitions in Methods), suggesting that the effect of implantation was negligible (Fig. S17).**

(Page 21 line 18- Page 22 line 1, **Methods section**) **The C57BL/6J mice were divided into the SHAM (sham surgery only without electrode implantation), sham (sham surgery and flexible electrode implantation without 2D ES), 2D ES (tendon surgery, flexible electrode implantation and 2D ES), non-ES (tendon injury and flexible electrode implantation without ES) and t-ES (tendon surgery, flexible electrode implantation and traditional commercial ES based on the Intan RHS2000 commercial stimulator) groups.**

The corresponding revisions in Supplementary Materials Section 17:

Section 17. The influence of implantation

Figure S17 | The level of IL-6 in blood before and after implantation. The result between SHAM and sham groups showed that implantation had a negligible effect on the elevation of inflammatory cytokines. ns, no significance.

Comment 12:

Add a reference to “fig 4e” in the corresponding sentence (page 12, line 15).

Response:

We thank the reviewer for this kind suggestion. We have **added a reference to Fig. 4e** and **additionally supplemented mitochondrial area as an indicator of neuron damage in Supplementary Figure S18.**

Neuronal death was estimated based on the expansion of mitochondria reflected on its size (**Ref 1**). We have evaluated the maximum diameter of mitochondria in the original manuscript, and further supplemented the area of mitochondrial as an indicator, with reference to previous reports (**Ref 1**). The statistics of mitochondrial area are shown in **Figure R10**, where **higher currents result in larger mitochondrial areas**, which is similar to the maximum diameter results, **suggesting that higher stimulation currents could cause damage to sympathetic neurons.**

(Figure S18 right panel) Figure R10. Area of mitochondrion among the 2D ES (~0.175 mA), t-ES (0.1, 0.3 mA) groups. The result showed that mitochondria in t-ES of 0.3 mA was expanded compared with 2D ES and t-ES of 0.1 mA, which implied that a higher current could cause damage to sympathetic neurons. ns, no significance; *, $p < 0.05$; ***, $P < 0.001$; ****, $p < 0.0001$.

References:

- Jiang S, Nandy P, Wang W, et al. Mfn2 ablation causes an oxidative stress response and eventual neuronal death in the hippocampus and cortex[J]. Molecular neurodegeneration, 2018, 13(1): 1-15. **(Newly added reference for Fig. 4e)**

The corresponding revisions in the revised manuscript:

(Page 13 line 17-20) In contrast, 0.1 mA is damage-free to neurons, similar to non-ES ones, which had been verified quantitatively by the maximum diameter and area of mitochondria (a strong indicator of cellular activity⁴⁴, Fig. 4e right panel, Fig. S18).

The corresponding revisions in Supplementary Materials Section 18:

Section 18. Low stimulation current caused little nerve damage

(Revised Figure S18) Figure S18 | Low stimulation current caused little damage on sympathetic neuron. Swelling rough endoplasmic reticulum and low electron density of mitochondrion in the 0.3 mA ES indicated that high current ES is damaging to sympathetic neurons. In contrast, the 0.1 mA ES and 2D ES caused little damage to neurons, like that of non-ES neurons. This suggests that 2D ES with lower stimulation current is expected to reduce nerve damage. Yellow arrow showed rough endoplasmic reticulum and red arrow with pink shadows showed mitochondrion. Area of mitochondrion was also counted. The result showed that mitochondria in t-ES of 0.3 mA was expanded compared with 2D ES and t-ES of 0.1 mA, which implied that a higher current could cause damage to sympathetic neurons. ns, no significance; *, $p < 0.05$; ***, $P < 0.001$; ****, $p < 0.0001$. Scale bar from left to right: 2, 1, 0.5 μm (orange lines).

Comment 13:

Specify the applied ES profile (t-ES or 2D ES) in figure Page 14 line 2 and figure S18.

Response:

We thank the reviewer for this kind reminder. For **Figure S25 (Figure S18 in origin version)** on the effect of blood flow velocity, we used **2D ES**. Laser speckle was performed on 2D ES group to assess the blood flow and showed that 2D ES had no influence on blood flow of hind paws. We have revised **Figure S25** and added the relevant description in the manuscript to specify 2D ES used.

The corresponding revisions in the revised manuscript:

(Page 15 line 14-17) Moreover, to rule out a potential anti-inflammation effect of **2D ES** through blood flow of hind paws, laser speckle was performed and showed no significant changes in blood flow velocity before, during and after **2D ES** with current in the threshold range (**Fig. S25**).

The corresponding revisions in Supplementary Materials Section 23:

Section 23. Determine the effects of 2D ES on blood flow

(Revised Figure S25) Figure S25 | 2D ES had no influence on blood flow of hind paws. Laser speckle was performed and illuminated that blood flow velocity had no significant change before, during and after 2D ES with the current allowed by the threshold range. ns, no significance.

Comment 14:

Provide more details about the tissue origin for IL-6 measurement in figure legends 4b and Methods section.

Response:

We thank the reviewer for this suggestion. **We have added the details of tissue origin in the caption of Figure 4b and Methods section.**

The tissue around the injury tendon is the origin for IL-6 measurement. Specifically, we first removed the skin on the sole of the foot, cut the tendon, and then **harvested all of the tissue containing muscle and connective tissue** (except for the bone tissue). Every tissue is approximately 20 mg. 100 µl of PBS was added to every tissue in a 1.6 ml EP tube and grinded using an ultrasonic grinder. The tissues were then centrifuged at 5000× g for 5 minutes at 4°C and the supernatant was collected for use in ELISA. The above description has been added in the revised manuscript.

The corresponding revisions in the revised manuscript:

(Page 22 line 19-22, **Methods section**) **ELISA.** The tissues around the tendon containing muscle and connective tissue were harvested for ELISAs. Every tissue was about 20 mg. 100 µl PBS was added to every tissue in the 1.6 ml EP tube and grinded using an ultrasonic grinder. Then the tissues were centrifuged at 5000× g for 5 minutes at 4°C and the supernatant was collected for ELISAs.

(Page 35 line 4-5, **Figure caption of Figure 4b**) **b**, The temporal variation of IL-6 in tissue around the injury tendon after surgery. ELISA test for IL-6 between sham and non-ES groups at 0, 1, 2, 3h after surgery.

Comment 15:

Table S1 need to be amended:

a. Microcuff electrodes were reported to be used by ref 5 for in vivo pancreatic nerve electrical stimulation not ‘hook electrodes’.

b. In this table what does ‘specificity’ exactly means in that case? In my view, there is no apparent rationale for considering the microcuff electrodes utilized in reference 5 to be less specific than employing an electrode to encase the sympathetic chain, as demonstrated in the present study.

c. Excluding ultrasounds from this table is advisable, given the challenging nature of comparing electrical and ultrasound strategies in terms of amplitude, stimulus site, signal properties, and specificity;

d. Please specify ‘real-time monitoring of nerve stimulation’ instead of ‘real-time monitoring’, which is confusing since it could refer to real-time monitoring of spontaneous sympathetic nerve activity.

Response:

We thank the reviewer for these valuable suggestions. **We have revised Table S1 as recommended, including correcting Ref 5 for electrode type (microcuff), revising the stimulation mode (replacing specificity with direct nerve stimulation), removing the ultrasound technique, and supplementing real-time monitoring of nerve stimulation.** Here, we divided the reviewer’s suggestions into four parts (in blue) and responded to each of them specifically (in black), and the revised **Table S1** is shown below.

a. Microcuff electrodes were reported to be used by ref 5 for in vivo pancreatic nerve electrical stimulation not ‘hook electrodes’.

Thanks to the reviewer for the kind reminder, **we had replaced “hook electrodes” with “microcuff electrodes” in the revised Table S1.**

b. In this table what does ‘specificity’ exactly means in that case? In my view, there is no apparent rationale for considering the microcuff electrodes utilized in reference 5 to be less specific than employing an electrode to encase the sympathetic chain, as demonstrated in the present study.

“Specificity” in the original manuscript meant that the ES was performed directly on the nerve. However, as mentioned by the reviewer in Comment 3, directly targeting the sympathetic chain is not sufficient for ES to be “specific”. Because the sympathetic chain contains not only sympathetic nerves, but also sensory nerves, albeit in small proportions. The description “Specificity” may not be appropriate because the potential influence of sensory nerves cannot be excluded. To more accurately describe the mode of ES, we have **replaced “Specificity” to “Direct nerve stimulation” to emphasize the fact that the flexible electrode wraps around the sympathetic nerve chain to perform direct ES** (as distinct from indirect stimulation with electroacupuncture or manual acupuncture, which takes place in the somatic tissues, where the stimulating needles are located away from the ganglion and **do not act directly on the nerve bundle**). And the ES mode of microcuff electrodes on pancreatic nerves in Ref 5 was also classified as direct nerve stimulation.

c. Excluding ultrasounds from this table is advisable, given the challenging nature of comparing electrical and ultrasound strategies in terms of amplitude, stimulus site, signal properties, and specificity;

We appreciate the reviewer’s suggestion that piezoelectric scaffolds performed by ultrasounds is to some extent difficult to compare with other electrical strategies. Therefore, **we have removed it in the revised Table S1.**

d. Please specify ‘real-time monitoring of nerve stimulation’ instead of ‘real-time monitoring’, which is confusing since it could refer to real-time monitoring of spontaneous sympathetic nerve activity.

We thank the reviewer for the suggestion. “Real-time monitoring” is ambiguous, which could refer to real-time monitoring of spontaneous sympathetic nerve activity. **We have replaced it with “real-time monitoring of nerve stimulation”.**

Revised Table S1 (Revisions are highlighted):

Electrostimulator system	Microcuff electrode	Electroacupuncture	Electroacupuncture	Commercial stimulators	2D Neuromorphic Electrostimulator
Targeted symptom (diseases)	Autoimmune diabetes	Systemic inflammation	Systemic inflammation	Inflammation in tendon injuries	Inflammation in tendon injuries
Stimulus site	Near pancreatic nerve	Acupoints (ST36, ST25)	Acupoints (ST36, ST25)	Sympathetic chain	Sympathetic chain
Direct nerve stimulation	Yes	No	No	Yes	Yes
Stimulus signal	Fixed rectangular charged-balanced biphasic pulses	Fixed rectangular charged-balanced biphasic pulses	Fixed rectangular charged-balanced biphasic pulses	Fixed rectangular charged-balanced biphasic pulses	Programmable bionic spikes
Current amplitude	0.45 mA	0.5-3 mA	0.5 mA	0.3-3 mA	average of ~0.175 mA (0.04-0.6 mA)
Real-time monitoring of nerve stimulation	incapable	incapable	incapable	incapable	capable
References	Nat. Biotechnol. 2019 [5]	Neuron 2020 [6]	Nature 2021 [7]	Our work	Our work

Comment 16:

Discuss the implications of the role of the β -2 adrenergic receptor on myeloid cells in light of Simon et al. 2023 (PMID: 37123375).

Response:

We thank the reviewer for this kind suggestion, and we have **added the reference recommended by the reviewer and related discussions of β -2 adrenergic receptor on myeloid cells to the revised manuscript.**

Simon et al. has reported that activation of the cholinergic anti-inflammatory pathway (CAP) by stimulation of the vagus or splenic nerves in mice is mainly mediated by direct binding of the NE to the β 2-AR on splanchnic macrophages (**Ref 1**).

Similarly, we have inferred that anti-inflammatory in our model by ES of SChG in mice is mainly mediated by directly binding of NE to the β 2-AR on macrophages.

References:

1. Simon T, Kirk J, Dolezalova N, et al. The cholinergic anti-inflammatory pathway inhibits inflammation without lymphocyte relay[J]. *Frontiers in Neuroscience*, 2023, 17: 1125492. **(Newly added reference)**

The corresponding revisions in the revised manuscript:

(Page 17 line 4-6) **Moreover, Simon et al. found that activation of the cholinergic anti-inflammatory pathway (CAP) by stimulation of vagus or splenic nerves in mice is mainly mediated by direct binding of NE to β 2-AR on splenic macrophages.**

Reviewer #3

General comments:

The paper presents a neuromorphic electrical stimulator based on a two-dimensional (2D) ultra-thin semiconductor floating-gate memory (FGM) interdigital structure of an electronic circuit. The 2D floating-gate memory devices are presented to be programmable to mimic progressive synaptic plasticity and fire neuron-like spikes for stimulation, thus claiming to minimize nerve damage. The paper demonstrates the isolation of the sympathetic chain in retroperitoneal space and delivers customized stimulation to the segment of the injured tendon by using a flexible electrode array. This helps in decreasing the inflammatory response due to cytokine protein. The paper further claims to adopt a low stimulus magnitude with an average current of 175 μA , which helps in minimizing damage to the sympathetic neurons. The authors also report that the proposed stimulation exerts anti-inflammation effects through macrophage $\beta 2$ adrenergic signaling by using mice subjects.

The paper is well-written and includes a wide range of measurements. I have personally liked the ideas of neuromorphic spike generation for stimulation, atomic-scale thin transistors with memory effects, and reduction of inflammation through low-magnitude stimulation. The diagrams are nice, and the paper seems thorough.

I have a few questions and concerns. The authors may consider addressing the following concerns.

Response:

We appreciate the reviewer's careful reading of our manuscript and **the positive comments** (such as "The paper is well-written and includes a wide range of measurements", "I have personally liked the ideas of neuromorphic spike generation for stimulation, atomic-scale thin transistors with memory effects, and reduction of inflammation through low-magnitude stimulation", "The diagrams are nice, and the paper seems thorough") **as well as constructive suggestions**. As summarized by the reviewer, we proposed a neuromorphic ES strategy based on MoS₂ floating-gate memory (FGM) interdigital circuit (IDC). Direct stimulation was achieved by wrapping

the sympathetic nerve chain with flexible electrodes, and MoS₂ FGM IDC was programmed to fire bionic spikes, thereby downregulating inflammatory cytokines with minimal nerve damage. Subsequently, we have used transgenic mice to further determine the biological mechanisms by which ES exerts its anti-inflammatory effects.

Regarding one of the major concerns about charge balancing mentioned by the reviewer, we have **added a new section in Supplementary Materials to discuss the charge balancing** in detail and **verified the charge balance of neuromorphic ES solution by calculating the charge quantities per stimulus pulse**. In addition, we have **discussed the advantages of memory effect-based stimulation** and described ways for removing this memory effect. We have responded the reviewer's concerns point by point, made corresponding discussions and revisions (highlighted in yellow) to the main text and Supplementary Materials. **A summary of the revisions in response to the reviewers' comments can be found at the end of the rebuttal letter.** We hope the revised manuscript could address all your concerns.

For detailed comments, the main responses are summarized as follows:

- Added a discussion of charge balancing in neuromorphic stimulation system to the **Supplementary Materials Section 12**;
- Supplemented a threshold current comparison between biphasic pulses (t-ES) and bionic spikes (2D ES), as well as calculated charge quantities per stimulus pulse, and relevant discussions had been added in **Supplementary Materials Section 20**;
- Supplemented the strength-duration (S-D) curve for the bionic stimulus, and the results showed that rheobase and chronaxie is ~39.69 pA, ~141.95 ms, respectively;
- Added discussions on the effect of different stimulation patterns, including rectangular, sinusoidal (exponential) and neuromorphic bionic waves;
- Discussed **the advantages of memory effect-based stimulation**, and described ways to remove this memory effect, if needed;
- Explained the reason for the 1 h waiting time after implantation, and **stated the focus of this work (inhibit inflammation from early acute injury)**;

- Explained the reason for the lower current in **Figure 3f, Figure S11** (magnitude of nA, **single-channel MoS₂ FGM**) than the claimed stimulation current and the stimulation current in **Table S1** (magnitude of μA , **multi-channel FGM IDC**);
- Discussed the safety of optogenetic stimulation (a non-electrical neurostimulation) and compares it with our proposed solution;
- **Renumbered the supplementary data** according to the order of their appearance in the main text, and updated the Supplementary Figures.
- Added a schematic diagram to **Figure S1** and revised the associated figure caption, revised and reorganized **Figure S6** and enlarged the captions for clear visibility;
- Supplemented the finite element simulation of FGM device with COMSOL Multiphysics in **Supplementary Materials Section 7** to fully show its working mechanism;
- Clarified the real-time monitoring of nerve stimulation **from the perspective of working strategy**, which is not available with existing stimulators;
- Bolded the scale bars in **Figure S18** to make them clearly visible, supplemented the blue and red curves of **Figure 3f** with time axes and added a discussion of time correlation (spike frequency).

Comment 1:

How charge balancing is accomplished while adopting the presented neuromorphic stimulation pulses? I believe that charge balancing a must-have criterion for any electrical stimulation of the nervous system, which has not been addressed in the manuscript.

Response:

Thank the reviewer for the valuable comment. In the revised version, **we have added a discussion of charge balancing in neuromorphic stimulation system to the Supplementary Materials Section 12**, as suggested by the reviewer.

We agree with the reviewer that charge balance is a major concern in functional electrical stimulation (ES), since **the charge accumulation over time could lead to electrolysis and cause tissue destruction (Ref 1)**. Similarly, **for our proposed neuromorphic electrical stimulation, it follows the charge balance criterion**. **Figure R11a** shows a comparison between existing stimulators and our proposed neuromorphic ES solution, where different working principles lead to different charge balance approaches: 1) For existing stimulators, the ES waveforms usually consist of a positive pulse and a negative pulse, **with equal integration of the positive and negative pulse currents in time (shaded portion of the figure)**, that is, **charge balance is achieved**, as shown in **Figure R11b**. Such biphasic waveforms are directly applied to the biological tissue/nerve (**top panel in Figure R11a**), and benefits from a charge balance that reduces tissue damage caused by electrolysis (**Ref 1**).

2) For neuromorphic ES, these typical biphasic charge-balanced waveforms are applied to the 2D MoS₂ floating-gate memory (FGM) interdigital circuit (IDC) through an analyzer, whose output bionic spikes are used as nerve stimulus (**bottom panel in Figure R11a**). **And the bionic spikes output by the neuromorphic 2D FGM IDC could be programmed for charge balancing**. **Figure R11c** shows a representative bionic spike generated by the neuromorphic device (extracted from **Figure 3f**). The dashed line can be considered as the baseline of the bioelectric current induced by intracellular ions. By integrating the relative positive and negative currents, it can be calculated that **both positive and negative charge in one period are $\sim 2.4 \times 10^{-12}$ C, which implies that charge balancing is reached**. In addition, for neuromorphic ES, once the charge balance at the input terminals is broken (due to variations in voltage pulses), the electrolytic process and charge accumulation will take place in the FGM IDC device rather than in the biological tissues/nerves, which can serve as **an additional protective effect**.

Based on the reviewer's suggestions, and to more fully describe functional stimulation, we have added a discussion of charge balancing for neuromorphic bionic spikes in both the revised manuscript and Supplementary Materials.

(Supplementary Figure S12) Figure R11. Charge balancing in functional electrical stimulation systems. a, Comparison of existing and neuromorphic ES solutions. For the existing ES, mouse tissues or nerves are connected to the stimulator via flexible electrodes, and the stimulator gives a charge-balanced biphasic square wave signal with a fixed stimulation amplitude; for the proposed neuromorphic ES, the mouse is connected to the 2D FGM IDC via flexible electrodes, and the analyzer applies voltage pulses to the FGM IDC to enable it to output programed bionic spikes as stimuli, while the analyzer monitors and displays the current of the stimulation loop in real time. **b**, Typical waveform of an existing stimulator. The waveform contains positive and negative current pulses that can be tuned in duration and amplitude to achieve charge balance. **c**, Representative bionic spike waveform in neuromorphic ES, which can be programmed for charge balancing.

References:

1. K. Sooksood, T. Stieglitz and M. Ortmanns, "Recent advances in charge balancing for functional electrical stimulation," 2009 Annual International Conference of the

The corresponding revisions in the revised manuscript:

(Page 10 line 14-Page 11 line 2) **Fig. 3f** presents the extracted intrinsic bioelectrical signals of sympathetic nerve (top panel, see **Fig. S10b** for detailed sympathetic nerve signals).....which provides potential to inhibit inflammation with lower stimulation amplitudes and minimal biological damage. Additionally, similar to existing stimulators, the bionic spikes generated by the FGM IDC allow for charge balancing, thereby minimizing tissue damage caused by charge accumulation and electrolysis (**Supplementary Section 12**).

Related discussions have been added in Supplementary Materials Section 13:

Section 12. Charge balancing in neuromorphic stimulation system

The proposed neuromorphic ES exhibit a different charge balancing approach than the existing ES due to different working principles, as shown in **Figure S12a**. For existing stimulators, the ES waveform typically consists of a positive pulse and a negative pulse, with the positive and negative pulse currents being integrated equally in time (shaded portion of the figure), i.e., charge balance is achieved, as shown in **Figure S12b**. This biphasic waveform acts directly on biological tissues/nerves (top panel of **Figure S12a**), and the charge balance reduces the tissue damage caused by electrolysis. While For neuromorphic ES, these typical biphasic charge-balance waveforms were applied via the analyzer to the 2D FGM IDC, whose output of bionic spikes was used as a nerve stimulus (**Figure S12a** bottom panel). The bionic spikes output from the 2D FGM IDC can be programmed for charge balancing. **Figure S12c** shows representative bionic spikes generated by the neuromorphic device (extracted from **Fig. 3f**). The dashed line can be considered as the baseline of the bioelectric current induced by intracellular ions. By integrating the relative positive and negative currents, it can be calculated that both positive and negative charge in one period is $\sim 2.4 \times 10^{-12}$ C, which means that charge equilibrium has been reached. Furthermore,

for neuromorphic ES, once the charge balance at the input terminals is broken (due to changes in voltage pulses), the electrolysis process and charge accumulation will take place in the FGM IDC device rather than in the biological tissues/nerves, which can serve as an additional protection.

Comment 2:

Can the authors show a detailed comparison between the threshold stimulus magnitude (or current) for both the bionic spike-like stimulus pulses and the standard biphasic pulses? What are the charge quantities per stimulus pulse while a specific gate bias voltage has been adopted?

Response:

We thank the reviewer for this valuable suggestion. We have **supplemented a threshold current comparison between biphasic pulses (t-ES) and bionic spikes (2D ES), as well as calculated charge quantities per stimulus pulse, and the relevant discussions have been added to the revised Supplementary Section 20.**

The average of IL-6 in non-ES group was taken as the baseline (~122 pg/mg prot), and IL-6 levels below the baseline after stimulation were defined as effective inhibition, with the corresponding stimulation current being the threshold. As shown in **Figure R11**, compared to the threshold of standard biphasic pulse generated by t-ES (~300 μA), the **bionic spikes generated by 2D ES features a lower threshold stimulus current of ~40 μA .**

In addition, we have **supplemented calculation of charge quantities for both standard and bionic stimulus spikes.** With a fixed gate bias, the integrals of stimulus currents over time (100 cycles) are calculated and averaged, which is defined as the charge quantities in a single stimulus pulse. Taking **Figure S13** as an example, **charge quantities per stimulus pulse for bionic spike is calculated to be $\sim 1.19 \times 10^{-6}$ C.** While for the standard biphasic pulse, we used a stimulus current of 300 μA , since it has a similar stimulation efficacy to the bionic spike pulse. **The calculated charge**

quantities per stimulus pulse for the standard biphasic pulse is $\sim 6 \times 10^{-6}$ C, which is much larger than that of the bionic stimulus spikes.

(Supplementary Figure S22) Figure R12. Threshold range of stimulation current for both t-ES and 2D ES. The IL-6 concentrations under the effect of different stimulation current amplitudes were counted (same stimulation frequency and duration of 10 Hz, 15 min), and the results showed that the 2D ES threshold for effective inflammation inhibition ranged from 40~600 µA (average amplitude of ~ 175 µA), while t-ES for effective inflammation inhibition ranged from 300~2000 µA. Inflammatory cytokines appear elevated in response to smaller and larger current amplitudes (baselines for IL-6 is ~ 122 pg/mg prot).

The corresponding revisions in the revised manuscript:

(Page 14 line 16-19) As shown in Fig. 4c red arrow, the 2D ES group with average currents of ~ 0.175 mA resulted in a further 70.6% drop in IL-6 levels compared to the t-ES group with fixed currents at 0.175 mA, which suggests that the direct neuromorphic bionic spikes lowered the threshold for ES efficacy (t-ES threshold is 0.3~2 mA).

Comment 3:

Can the authors show the strength duration curve for the bionic stimulus waves?
Can you state the measured chronaxie and rheobase?

Response:

We thank the reviewer for this comment. We have **supplemented the strength-duration (S-D) curve for the bionic stimulus**, and the results showed that **rheobase and chronaxie is ~39.69 pA, ~141.95 ms, respectively.**

The response to functional ES is directly related to the amplitude and duration of the stimulus current to which the neuron is exposed. This threshold current is nonlinearly dependent on the duration of the depolarizing phase of ES. **The relationship between neural threshold current and stimulus duration can be described by the S-D curve**, which plots stimulus pulse width versus threshold current (**Ref 1**). To explore the S-D curve for the bionic stimulus waves, we delivered bionic stimulus to individual neuron on mouse brain slides and collect its action potential. We used 2D FGM IDC to deliver bionic stimulation of individual neurons on mouse brain slices and collected their action potentials by patch clamp (**Ref 2**). We have applied pulse widths of 50, 100, 200, 500, 1000, and 2000 ms, with stimulation starting at 0 pA in an incremental step of 5 pA (output stimulus current of 2D FGM IDC can be flexibly programmed to the desired value), until the action potential of neuron was elicited. Then the S-D curve for bionic stimulus was obtained, as shown in **Figure R13**, where the black dashed line is the fitting curve. It was determined that the **rheobase (I_r , minimum activation current required for an infinite duration pulse)** is ~39.69 pA and the **chronaxie (C , required pulse width at 2 times the rheobase)** is ~141.95 ms.

Figure R13. The strength duration curve for the bionic stimulus waves. The individual neuron on mouse brain slides was delivered bionic stimulus and its action potential was collected. Pulse width was set at 50, 100, 200, 500, 1000, and 2000 ms and started the stimulus from 0 pA with an incremental step of 5 pA until the action potential was elicited. The black dotted line is the fitting curve. I_r : rheobase, C : chronaxie.

References:

1. Colombo J, Parkins C W. A model of electrical excitation of the mammalian auditory-nerve neuron[J]. Hearing research, 1987, 31(3): 287-311.
2. Pelot N A, Grill W M. In vivo quantification of excitation and kilohertz frequency block of the rat vagus nerve[J]. Journal of neural engineering, 2020, 17(2): 026005.

Comment 4:

How do you compare presented neuromorphic stimulus pulses with alternate stimulus patterns, such as exponential and other waves?

Response:

We thank the reviewer for this comment. **In the revised manuscript, we have added discussions on the effect of different stimulation patterns.**

In our manuscript, **we focus on comparing neuromorphic stimulation patterns with biphasic rectangular stimulation patterns.** Currently, functional electrical stimulation uses a number of other stimulation patterns, such as sinusoidal or

exponential waves. **However, among all these patterns, rectangular waves are most widely used in clinical and commercial electrical stimulation, especially for neurostimulation**, which is of interest to us (Ref 1, 2). On the one hand, rectangular waves are effective in activating nerve cells due to their fast transition edges. It has been proved that compared to other waves (sinusoidal, exponential or triangular waves), **rectangular waves have a lower chronaxie and similar rheobase, suggesting a high stimulation efficiency (Ref 3)**. By contrast, sinusoidal or exponential waves own other characteristics, which make them suitable for treating different symptoms. According to previous studies, these waves **have the advantage of long-term stimulation due to their lower energy consumption (Ref 4)**. In addition, waveforms have been reported to affect the induced sensation. While sinusoidal or exponential waves **show a relatively lower rate of current variation, making stimulation more natural and adaptable**. These waves could also result in larger sensation areas, indicating that more fibers are being recruited (Ref 3), which **may be useful for muscle relaxation**.

As for the presented neuromorphic bionic spikes, they have **a rapid rising edge of ~1 ms** (extracted from **Figure 3f**), which **enables efficient activation of nerve cells**, similar to rectangular waves. In addition, bionic spikes show **better inflammation inhibition efficacy** than rectangular waves (more inflammatory factor decline with lower stimulus current). It is worth noting that since rectangular waves have a similar rheobase and lower chronaxie than other waveforms (Ref 3), comparisons with rectangular waves may reflect conclusions with other waveforms to some extent.

References:

1. Grill W M. Model-based analysis and design of waveforms for efficient neural stimulation[J]. Progress in brain research, 2015, 222: 147-162.
2. Imatz Ojanguren, E. (2019). Functional Electrical Stimulation. In: Neuro-fuzzy Modeling of Multi-field Surface Neuroprostheses for Hand Grasping. Springer Theses. Springer, Cham.
3. Collu, R., Earley, E.J., Barbaro, M. et al. Non-rectangular neurostimulation waveforms elicit varied sensation quality and perceptive fields on the hand. Sci Rep 13, 1588 (2023).

4. Foutz TJ, McIntyre CC. Evaluation of novel stimulus waveforms for deep brain stimulation. J Neural Eng. 2010 Dec;7(6):066008.

The corresponding revisions in the revised manuscript:

(Page 10 line 9-14) Specifically, MoS₂ FGM IDC stimulator is capable of generating bionic spikes, while commercial stimulators (such as Intan RHS2000) generally provide a fixed, abrupt biphasic current pulses (Fig. S10a). Although functional electrical stimulation may also employ sinusoidal or exponential waves for inducing sensation, muscle relaxation, etc., for nerve electrical stimulation, we focus on rectangular waves for comparison, considering the high efficiency of rectangular waves for activating nerve cells.

Comment 5:

The reported stimulus pulses at various gate bias voltages in Figure S11 show a memory effect and a non-uniform pulse height over time. What are the specific advantages of memory effect-based stimulation? How can you remove such a memory effect in the future if such an effect is required to be ignored?

Response:

Thank the reviewer for the valuable comments. Advantages of memory effect-based stimulation mainly include the ability **to achieve intrinsic neuromorphic bionic spikes**, which **effectively lowers stimulus current amplitude and damage**, as well as the ability **to greatly enhance stimulus programmability**. And **this memory effect can be eliminated by removing the floating-gate structure (potential well) of the stacked Al₂O₃/HfO₂/Al₂O₃ (A/H/A) dielectric**. In the following, we divided the reviewer's questions into two parts (in blue) and answered each in detail (in black), and we have supplemented the relevant discussions in the revised manuscript.

“What are the specific advantages of memory effect-based stimulation?”

As mentioned by the reviewer, **Figure S11** shows typical stimulus pulse waveforms at different gate biases with a memory effect. To be specific, **we introduced stacked A/H/A dielectric as a floating-gate in the device structure to realize the memory effect**, which consists of a tunneling layer, a trapping layer, and a blocking layer, as shown in **Figure R14a**. Here we illustrate the memory effect with an example of write operation (**Figure R14b**): electrons in the atomically thin semiconductor MoS₂ channel tunnel into the A/H/A floating gate under the external gate electric field (positive gate pulse voltage), which corresponds to a decrease in channel current/conductance and the device is in the high resistance state (HRS), and vice versa (low resistance state, LRS, **Figure R14c**). Since the stacked A/H/A dielectric introduces a potential well, the tunneled electrons are persistently trapped, resulting in a memory retention effect (**Ref 1**). However, despite the presence of the potential well, **there is still a relaxation effect of tunneled electrons in the trapping layer (Ref 2)**, and the current will slowly converge to the baseline. It is worth noting that the relaxation time is tunable and dependent on the thickness of the tunneling layer. By designing an appropriate relaxation time (**Ref 3**), **the memory effect-based stimulation is able to generate a stimulus spike similar to an action potential**, as shown in **Figure R15a**.

That is, a major advantage of floating-gate memory based on atomically thin MoS₂ is the ability to **generating intrinsic neuromorphic bionic spikes**, whereas other technologies, such as CMOS, usually require the additional introduction of specialized functional circuit blocks. And neuromorphic stimulation with bionic spikes **promises therapeutic benefits at lower stimulus current amplitude and reduced biological damage (Ref 4)**, and it has shown great success when acting as an afferent or efferent nerve for muscle contraction and limb movement (**Ref 5, 6**).

In addition, another major advantage of memory effect-based stimulation is **the excellent programmability**. By controlling the write/erase process of the electrons in the floating gate or the external gate voltage bias modulation (**Figure R16, Figure S11**),

the stimulus pulse can be flexibly programmed, e.g., to mimic progressive synaptic potentiation/depression plasticity, as shown in **Figure R15b**.

“How can you remove such a memory effect in the future if such an effect is required to be ignored?”

As discussed above, we realized the memory effect by introducing the stacked A/H/A dielectric (including tunneling, trapping and blocking layers) as a floating-gate in the device structure (**Figure R14a**). **The elaborate stacked A/H/A floating gate introduces a potential well**, which keeps the tunneled electrons persistently captured and prevents them from returning to MoS₂ channel, thus resulting in a memory effect (**Ref 1**). The switching of memory state is achieved by writing (HRS) or erasing (LRS) electrons to the floating gate, as displayed in **Figure R14b, c**, respectively. **This memory effect can be eliminated by removing the stacked A/H/A floating gate**. Because when the stacked A/H/A floating-gate is removed, there is no longer a potential well, the tunneled electrons cannot be captured durably and will quickly return to the channel (**Ref 2, 3**), so that the memory effect no longer exists either.

Figure R14 | Energy band of 2D FGM. **a**, Energy bands of the different materials composing the 2D FGM prior to contact. E_c and E_v are the positions of the bottom of the MoS_2 conduction band and the top of the valence band, respectively. Stacked $AlO_x/HfO_x/AlO_x$ sequentially acts as a blocking, trapping and tunneling layer. **b**, 2D FGM is programmed by applying a positive gate voltage to inject electrons into the HfO_x (floating gate), thus enabling the HRS. **c**, Erase operation to extract electrons from the floating gate with negative gate voltage applied, thus switching back to LRS.

(Figure 3d, 3f) Figure R15 | Typical bionic stimulus spikes and their plasticity. a, Sympathetic nerve signals (top panel) and bionic stimulus spikes (bottom panel) emitted by the MoS₂ FGM-based neuromorphic stimulator. **b,** Device conductance shows a progressive increase and decrease, which mimics the potentiation/depression plasticity in biological synapses and enables programmable stimuli.

(Figure S11) Figure R16 | bionic stimulus spikes under gate base voltage variation. a~d, Bionic stimulus currents generated at gate base voltages of -2, -3, -4, and -5 V,

respectively, with pulse amplitude and width fixed at +15 V for 100 ns, pulse number of 50 and drain voltage of 1 V. This shows that the bionic spikes emitted by the neuromorphic electrostimulator can be modulated by the gate base voltage.

References:

1. Liu L, Liu C, Jiang L, et al. Ultrafast non-volatile flash memory based on van der Waals heterostructures[J]. Nature nanotechnology, 2021, 16(8): 874-881.
2. Liu C, Yan X, Song X, et al. A semi-floating gate memory based on van der Waals heterostructures for quasi-non-volatile applications[J]. Nature nanotechnology, 2018, 13(5): 404-410.
3. Chen H, Liu C, Wu Z, et al. Time-tailoring van der Waals heterostructures for human memory system programming[J]. Advanced Science, 2019, 6(20): 1901072.
4. Shannon R V. A model of safe levels for electrical stimulation[J]. IEEE Transactions on biomedical engineering, 1992, 39(4): 424-426.
5. Kim Y, Chortos A, Xu W, et al. A bioinspired flexible organic artificial afferent nerve[J]. Science, 2018, 360(6392): 998-1003.
6. Lee Y, Liu Y, Seo D G, et al. A low-power stretchable neuromorphic nerve with proprioceptive feedback[J]. Nature Biomedical Engineering, 2023, 7(4): 511-519.

Related discussions have been added to the revised manuscript:

(Page 9 line 22- Page 10 line 1) **The advantage of memory effect-based stimulation lies mainly in the greatly improved programmability, since 2D MoS₂ FGM IDC exhibits flexible biomimetic programmability.**

(Page 10 line 7-11) **Another major advantage of memory effect is the realization of intrinsic neuromorphic bionic stimulation pulses, which effectively lower the stimulation current amplitude and biological damage. Specifically, MoS₂ FGM IDC stimulator is capable of generating bionic spikes, while commercial stimulators (such as Intan RHS2000) generally provide a fixed, abrupt biphasic current pulses (Fig. S10a).**

(Page 11 line 5-6) **It is worth noting that this memory effect can be eliminated by removing the stacked A/H/A floating gate.**

Comment 6:

Electrode implantation in retroperitoneal space has been done for 1 hour only, which sounds inadequately short for assessing long-term effectiveness. Please justify the choice of this time duration and how it strongly correlates with long-term assessment.

Response:

We thank the reviewer for the comments on timing of electrode implantation and long-term assessment. The experimental strategy model is shown in **Figure R17a (Figure 4a bottom panel)**, with a 1 hour wait for surgery after electrode implantation. **There are two reasons for choosing this time duration:** (1) During implantation, we pull the hook to avoid abdominal vessels and organs to expose the sympathetic ganglia, which would compress the abdominal aorta. **It is critical to wait 1 hour after implantation, because this eliminates the blood-flow change of hind paws caused by the compressed blood vessels;** (2) As shown in **Figure R17b (Figure 4b)**, **electrode implantation had no significant effect on the elevation of inflammatory factors** (the sham group is defined as sham surgery and flexible electrode implantation without ES, while the non-ES group is defined as tendon surgery and flexible electrode implantation without ES), so we **did not have to preempt implantation by about 1 week as in some reported work (Ref 1)**.

When it comes to long-term assessment, **the animal model in this work focused on early acute injury**, and acute inflammatory indicators were tested 2 hours after tendon surgery, so **long-term effectiveness of ES was outside the scope of this work**. Nevertheless, long-term electrical stimulation for the treatment of chronic disease is also a matter of concern, which we will work on in our future work.

Reference:

1. Guyot M, Simon T, Ceppo F, Panzolini C, Guyon A, Lavergne J, Murriss E, Daoudlarian D, Brusini R, Zarif H, Abélanet S, Hugues-Ascery S, Divoux JL,

Lewis SJ, Sridhar A, Glaichenhaus N, Blancou P. Pancreatic nerve electrostimulation inhibits recent-onset autoimmune diabetes. *Nat Biotechnol.* 2019 Dec;37(12):1446-1451.

(Figure 4a bottom panel, 4b) **Figure R17. a**, The anatomy of sympathetic trunk and ES strategy model. **b**, The temporal variation of IL-6 in tissue around the injury tendon after surgery. ELISA test for IL-6 between sham and non-ES groups at 0, 1, 2, 3 h after surgery. And IL-6 increased significantly at 2 h after surgery, which provided the timepoints of ES. $n = 3$. ns, no significance; *, $p < 0.05$; ****, $p < 0.0001$.

Comment 7:

The paper claims to have a low electrical stimulus current (175 μA average). However, the waveforms in Figure S11 show only nA currents.

Response:

We thank the reviewer for the comment on current magnitude. **Figure S11** shows the current tunability of the **single-channel MoS₂ FGM** with the device structure shown in the **left panel of Figure R18a**. While the neuromorphic electrostimulator for inflammation inhibition is **composed of multi-channel FGM IDC**, as shown in the **right panel of Figure R18a**.

With the accumulation of interdigital channels, the current could increase significantly (Ref 1-4) and can reach the order of (tens~hundreds) μA or even mA to meet the stimulation demand (Figure R18b), as we have discussed in the original

version. **Figure R18c** shows a representative stimulation current, which is in the order of hundred μA . To avoid potential misinterpretation and confusion, we have **added relevant explanations of stimulus current magnitude in the revised version.**

(Figure 2d, Figure S3b, Figure S13a, c) Figure R18 | Schematic of MoS₂ FGM IDC and current climbing effect. a, Structure of single-channel (left) and multi-channel (right) MoS₂ FGM IDC with blue, red and yellow dashed boxes representing the MoS₂ channel, control gate and interdigital electrodes, respectively. **b,** Current climbing effect of MoS₂ FGM IDC, where the output current of FGM can increase up to the mA level with the accumulation of interdigital channels. **c,** Representative bionic spikes with a stimulus amplitude of $\sim 200 \mu\text{A}$ and a frequency of 10 Hz.

References:

1. Yun B K, Kim H S, Ko Y J, et al. Interdigital electrode based triboelectric nanogenerator for effective energy harvesting from water[J]. Nano Energy, 2017, 36: 233-240.

2. Hu H, Pei Z, Fan H, et al. 3D interdigital Au/MnO₂/Au stacked hybrid electrodes for on-chip microsupercapacitors[J]. *Small*, 2016, 12(22): 3059-3069.
3. Beidaghi M, Wang C. Micro-supercapacitors based on interdigital electrodes of reduced graphene oxide and carbon nanotube composites with ultrahigh power handling performance[J]. *Advanced Functional Materials*, 2012, 22(21): 4501-4510.
4. Chen Y, Zhu C, Cao M, et al. Photoresponse of SnO₂ nanobelts grown in situ on interdigital electrodes[J]. *Nanotechnology*, 2007, 18(28): 285502.

Related revisions have been added to the revised manuscript:

(Page 9 line 2-3) **Fig. 3** depicts the unit performance of a single-channel 2D FGM in the neuromorphic ES IDC.

(Page 10 line 14-18) **Fig. 3f** presents the extracted intrinsic bioelectrical signals of sympathetic nerve (top panel, see **Fig. S10b** for detailed sympathetic nerve signals) and the bionic spikes (bottom panel) generated by a single-channel 2D FGM IDC under ultrafast gate pulses (see **Fig. S11** for bionic spikes under different gate base voltages).

(Page 11 line 2-5) And the neuromorphic electrostimulator consists of multi-channel MoS₂ FGM IDC. With the accumulation of interdigital channels, the increased current in the 2D FGM can meet the stimulation demand (the climbing of the stimulation current is shown in **Fig. S13a**) and act as a neuromorphic electrostimulator for inflammation inhibition.

Comment 8:

Likewise, what is the correlation between Fig. 3f and Table S1? The current magnitudes are grossly different, although they represent the bionic spikes.

Response:

We thank the reviewer again for raising questions about the current magnitude. Similar to discussed in Comment 7, the correlation between **Figure 3f** and **Table S1** is

that the former (**Figure 3f**) are bionic spikes emitted by a **single-channel MoS₂ FGM** (left panel of **Figure R18a**), while the bionic spikes of the latter (**Table S1**) are generated by the neuromorphic electrostimulator **with accumulated multi-channel FGM IDC** (right panel of **Figure R18a**).

As the interdigital channels accumulate, the **bionic spike current would increase significantly (Ref 1-4)**, which can reach the order of (tens~hundreds) μA to meet the stimulation demand (**Figure R18b, c**). To avoid potential misinterpretation and confusion, we have **added relevant explanations of spike current magnitude in the revised version**.

References:

1. Yun B K, Kim H S, Ko Y J, et al. Interdigital electrode based triboelectric nanogenerator for effective energy harvesting from water[J]. Nano Energy, 2017, 36: 233-240.
2. Hu H, Pei Z, Fan H, et al. 3D interdigital Au/MnO₂/Au stacked hybrid electrodes for on-chip microsupercapacitors[J]. Small, 2016, 12(22): 3059-3069.
3. Beidaghi M, Wang C. Micro-supercapacitors based on interdigital electrodes of reduced graphene oxide and carbon nanotube composites with ultrahigh power handling performance[J]. Advanced Functional Materials, 2012, 22(21): 4501-4510.
4. Chen Y, Zhu C, Cao M, et al. Photoresponse of SnO₂ nanobelts grown in situ on interdigital electrodes[J]. Nanotechnology, 2007, 18(28): 285502.

Related revisions have been added to the revised manuscript:

(Page 9 line 2-3) **Fig. 3** depicts the unit performance of a single-channel 2D FGM in the neuromorphic ES IDC.

(Page 10 line 14-18) **Fig. 3f** presents the extracted intrinsic bioelectrical signals of sympathetic nerve (top panel, see **Fig. S10b** for detailed sympathetic nerve signals) and the bionic spikes (bottom panel) generated by a single-channel 2D FGM IDC under ultrafast gate pulses (see **Fig. S11** for bionic spikes under different gate base voltages).

(Page 11 line 2-5) **And the neuromorphic electrostimulator consists of multi-channel MoS₂ FGM IDC.** With the accumulation of interdigital channels, the increased current in the 2D FGM can meet the stimulation demand (the climbing of the stimulation current is shown in **Fig. S13a**) and act as a neuromorphic electrostimulator for inflammation inhibition.

(Page 15 line 18-Page 16 line 2) Thanks to the bionic spikes as well as nerve-direct stimulation, the proposed MoS₂ FGM IDC-based neuromorphic 2D ES achieved greater efficacy of damage-free inflammatory inhibition (A performance benchmarks of the neuromorphic 2D ES versus existing stimulators are shown in **Table S1**, where bionic spikes refer to the range of spiking currents emitted by the multi-channel FGM IDC-based neuromorphic electrostimulator that are effective in reducing inflammatory cytokines).

Comment 9:

Optogenetic stimulus pulses are free from the charge delivery threshold and electrical stimulus artifacts? Can we consider such a non-electrical neural stimulation paradigm as a safer method than the reported stimulation scheme?

Response:

We appreciate the reviewer's valuable comments. Optogenetics has been widely used in neuroscience and it typically consist of two components: **photosensitized calcium channels on specific cells and optogenetic stimulus pulses**. The former is achieved by transgenic mice (e.g., Ai32(RCL-ChR2(H134R)/EYFP)) or adeno-associated viruses (AAVs) with genetically encoded Ca²⁺ indicators (GECIs) (e.g., GCaMP), while the latter was achieved by fiber-optic connected lasers (similar to flexible electrode-connected neuromorphic stimulator in our solution). However, **both transgenic mice and AVVs have been reported to be neurotoxic and will induce**

epilepsy (Ref 1). Therefore, **it is premature to consider optogenetic stimulation as a safer solution** than electrical stimulation. Since electrical stimulation only requires attention to implantation and stimulation parameters, there is no need to worry about virus issues. And **the proposed solution uses low-invasive implantation** (using the sympathetic ganglion anatomy located in the retroperitoneal space for low-invasive implantation), and the **neuromorphic bionic stimulation current does not cause damage to the target nerve.**

References:

1. Daigle T L, Madisen L, Hage T A, et al. A suite of transgenic driver and reporter mouse lines with enhanced brain-cell-type targeting and functionality[J]. Cell, 2018, 174(2): 465-480. e22.

*******Other Comments*******

Comment 10:

Fig S4 is a set-up photo and not a schematic.

Response:

We appreciate the reviewer's kind comment. In the revised version, we have **added a schematic diagram to Figure S1** (the original **Figure S4**, since the **supplementary figures were renumbered** according to the order of appearance in the main text), as shown in **Figure R19**, and **revised the associated figure caption** to make our architecture clearer.

(Revised Figure S1) Figure R19 | Set-up and schematic of the neuromorphic electrostimulator system. The core components of the proposed neuromorphic electrostimulator system include 2D MoS₂ FGM IDC stimulator with a scalable length of implanted electrodes, the FPC, the adapter PCB, the switch box and the analyzer/monitor.

The corresponding revision in the revised Supplementary Materials:

(Caption of Figure S4) Set-up and schematic of the neuromorphic electrostimulator system. The core components of the proposed neuromorphic electrostimulator system include 2D MoS₂ FGM IDC stimulator with a scalable length of implanted electrodes, the FPC, the adapter PCB, the switch box and the analyzer/monitor.

Comment 11:

Fig. S6: The captions are too small and unreadable. May be increased appropriately.

Response:

We thank the reviewer for this kind suggestion. In the revised version, **we have revised and reorganized Figure S6 and enlarged the captions for clear visibility, as shown in Figure R20 below.**

(Revised Figure S6) Figure R20 | Application of pulses with different widths. Fig. S6 shows pulses with different pulse widths from 100 ns to 100 ms, which corresponds to the actual pulses applied in the operating speed of **Fig. 3b** in the main text.

Comment 12:

Fig. S7 seems to be manually drawn. Can you show a simulation result for the same?

Response:

We thank the reviewer for this valuable suggestion on mechanism simulation. In addition to the original energy band analyses, we have **supplemented the finite element simulation of the device with COMSOL Multiphysics** to fully show its working mechanism in the revised **Supplementary Materials Section 7**.

Finite element simulation of the device is consistent with previous energy band analyses. **Figure R21d-f** shows the simulation results of the bandgap change of the 2D FGM in different states. When the gate voltage is fixed at 0V, the device lies in the flat band state (corresponding to **Figure R21a, d**), the conductive band and valence band shows almost no bending. When a positive pulse is applied to the gate (**Figure R21g**), the 2D FGM device turned into program state (**Figure R21b, e**). According to our simulation, both the conductive and valence band decreases near the surface, leading to a downward bend of the energy band. The electron moves across the tunneling layer and trapped by the floating gate, leading to an increase on both tunneling current and trapped charge. (see **Figure R21h, i**). While a negative pulse is applied, the 2D FGM device turned into erase state (**Figure R21c, f**). the energy bands increase and form an upward bend, removing the trapped charge and causing a negative tunneling current (see **Figure R21h, i**).

In the revised version, we have supplemented those simulation results in **Supplementary Materials Section 7** to make the working mechanism of 2D FGM more comprehensive.

(Revised Figure S7) Figure R21 | Energy band of 2D FGM. **a**, Energy bands of the different materials composing the 2D FGM prior to contact. E_c and E_v are the positions of the bottom of the MoS₂ conduction band and the top of the valence band, respectively. Stacked AlO_x/HfO_x/AlO_x sequentially acts as a blocking, trapping and tunneling layer. **b**, 2D FGM is programmed by applying a positive gate voltage to inject electrons into the HfO_x (floating gate), thus enabling the HRS. **c**, Erase operation to extract electrons from the floating gate with negative gate voltage applied, thus switching back to LRS. **d**, Simulated energy band of 2D FGM in flat band state. **e**, The simulated energy band of 2D FGM in program state. A downward bend energy band could be observed, which is consistent with panel b. **f**, Simulated energy band of 2D FGM in erase state. **g**, The voltage pulse used in program and erase. **h**, The tunneling current in program and erase, which indicates the electron's moving through barrier layer. **i**, The trapped charge in the floating gate.

Comment 13:

Fig. S19: Scale bars are too small and almost invisible.

Response:

We appreciate the reviewer's kind comment. In the revised version, we have **bolded the scale bars in Figure S18** (the original **Figure S19**) to make them clearly visible, as shown in **Figure R22** below.

(Revised Figure S18 left panel) Figure R22 | Low stimulation current caused little damage on sympathetic neuron. Swelling rough endoplasmic reticulum and low electron density of mitochondrion in the 0.3 mA ES indicated that high current ES is damaging to sympathetic neurons. In contrast, the 0.1 mA ES and 2D ES caused little damage to neurons, like that of non-ES neurons. This suggests that 2D ES with lower stimulation current is expected to reduce nerve damage. Yellow arrow showed rough endoplasmic reticulum and red arrow with pink shadows showed mitochondrion. **Scale bar from left to right: 2, 1, 0.5 μm (orange lines).**

Comment 14:

Table S1: Why do you claim that real-time monitoring (of nerve stimulation) is incapable with other stimulators.

Response:

We thank the reviewer for this valuable question. **A different working strategy from other (commercial) stimulators gives the proposed neuromorphic stimulator system the unique capability for real-time monitoring of the nerve stimulation loop current.**

Specifically, as shown in the **top blue panel of Figure R23**, existing commercial stimulators (e.g., Intan RHS2000) typically deliver fixed, abrupt (square) biphasic current pulses, which act as a “black box”, always **guaranteeing a preset fixed current amplitude (several to tens of mA), but no real-time current status of the stimulation loop is available (Ref 1-3)**. For the proposed neuromorphic stimulation system (**bottom red panel of Figure R23**), with the assistance of peripheral FPC, and adapter boards, 2D FGM IDC emits **programmable bionic spikes** for nerve stimulation in response to voltage pulses given by the analyzer (Keysight B1500A). **Simultaneously, the analyzer displays the loop current of 2D FGM IDC in real time, i.e., real-time monitoring of nerve stimulation and providing feedback on the target nerve**, which is not possible with other existing stimulators. To clarify the real-

time monitoring capabilities not available with existing stimulators and avoid potential ambiguity in the interpretation, we have added a discussion of real-time nerve stimulation monitoring in the revised version.

(Supplementary Figure S12a) Figure R23 | Comparison of existing and neuromorphic ES solutions. For the existing ES, mouse tissues or nerves are connected to the stimulator via flexible electrodes, and the stimulator gives a charge-balanced biphasic square wave signal with a fixed stimulation amplitude; for the proposed neuromorphic ES, the mouse is connected to the 2D FGM IDC via flexible electrodes, and the analyzer applies voltage pulses to the FGM IDC to enable it to output programed bionic spikes as stimuli, while the analyzer monitors and displays the current of the stimulation loop in real time.

References:

1. Liu S, Wang Z, Su Y, et al. A neuroanatomical basis for electroacupuncture to drive the vagal–adrenal axis[J]. *Nature*, 2021, 598(7882): 641-645.
2. Liu S, Wang Z F, Su Y S, et al. Somatotopic organization and intensity dependence in driving distinct NPY-expressing sympathetic pathways by electroacupuncture[J]. *Neuron*, 2020, 108(3): 436-450. e7.
3. Guyot M, Simon T, Ceppo F, et al. Pancreatic nerve electrostimulation inhibits recent-onset autoimmune diabetes[J]. *Nature biotechnology*, 2019, 37(12): 1446-1451.

Related discussions have been added to the revised manuscript:

(Page 11 line 6-13) With the assistance of peripheral FPC and adapter board, 2D FGM IDC can emit programmable bionic spikes for nerve stimulation in response to voltage pulses from the analyzer. The analyzer simultaneously displays the loop current of 2D FGM IDC in real time, i.e., real-time monitoring of nerve stimulation and providing feedback on the target nerve (see Fig. S13b, c for a typical real-time loop current at a stimulation frequency of 10 Hz and a duration of 15 min), while existing stimulators focus on settings that guarantee a fixed stimulus current amplitude and fail to monitor the current status in real time.

Comment 15:

Fig. 3f: The time axis is not labeled. Furthermore, is there a time correlation in between the blue and the red curves in this figure?

Response:

We thank the reviewer for this kind reminder. In the revised version, we have **supplemented the blue and red curves of Figure 3f with time axes and added a discussion of time correlation (spike frequency).**

The blue and red curves in **Figure 3f** show the intrinsic bioelectrical signals from sympathetic nerve (blue) and the bionic spikes generated by a single-channel MoS₂ FGM IDC under gate pulses, for which we have supplemented the time axes respectively, as shown in **Figure R24a**.

For the time correlation of spiking signals, **the main focus is on the frequency.** Since **Figure 3** illustrates the performance of the device cell (single-channel FGM IDC), for consistency reasons, the red curve in **Figure 3f** also shows the bionic spikes emitted by the single-channel device. Thus, **there is no direct time (frequency) correlation** between the sympathetic bioelectric signals and the bionic spikes of single-channel device demonstrated in **Figure 3f**. However, **the frequency of bionic spikes can be**

flexibly regulated to match biological signals or for targeted therapies. As shown in **Figure R24b**, this work used another representative bionic spike with a stimulus amplitude of $\sim 200 \mu\text{A}$ and a frequency of 10 Hz, which is expected to be effective in inhibiting inflammation (**Ref 1, 2**). To avoid any possible misinterpretation, for the bionic spikes, we have added a discussion of time correlation and frequency tunability in the revised manuscript.

(Revised Figure 3f, Figure S13c) Figure R24 | Typical sympathetic signals and bionic stimulus spikes. a, Sympathetic nerve signals (top panel) and bionic stimulus spikes (bottom panel) emitted by the MoS₂ FGM-based neuromorphic stimulator. **b**, Representative bionic spikes with a stimulus amplitude of $\sim 200 \mu\text{A}$ and a frequency of 10 Hz.

References:

1. Liu S, Wang Z F, Su Y S, et al. Somatotopic organization and intensity dependence in driving distinct NPY-expressing sympathetic pathways by electroacupuncture[J]. Neuron, 2020, 108(3): 436-450. e7.
2. Guyot M, Simon T, Ceppo F, et al. Pancreatic nerve electrostimulation inhibits recent-onset autoimmune diabetes[J]. Nature biotechnology, 2019, 37(12): 1446-1451.

Related revisions have been added to the revised manuscript:

(Page 10 line 14-21) **Fig. 3f** presents the extracted intrinsic bioelectrical signals of sympathetic nerve.....see **Fig. S11** for bionic spikes under different gate base voltages).

Bionic spikes are similar to bioelectrical signals, and although not directly time-correlated, the frequency of bionic spikes can be flexibly regulated to match biological signals or targeted therapies, which provides potential to inhibit inflammation with lower stimulation amplitudes and minimal biological damage.

Based on the reviewers' comments, we have performed the following revisions in the revised main text:

1. Page 1-2, we have replaced “specific” with “direct” and “macrophages” with “myeloid cell lineage (monocytes/macrophages and granulocytes)” throughout the manuscript to provide an accurate and comprehensive description of our results;
2. Page 3, we have added the comparison between electro-acupuncture (manual acupuncture) and the proposed neuromorphic ES solution;
3. Page 4, we have replaced Ref 18 on splenic nerve and added the discussion of sympathetic nerve, revised the abbreviation of sympathetic chain ganglia from “SchG” to “SChG” in the whole manuscript and corrected the typo of “lumbar”;
4. Page 7, we have corrected the figure number to show the position of cross-section scanning TEM image on the physical layout;
5. Page 9, we have revised the description of 2D FGM, added the advantages of memory effect-based stimulation and the reason for using rectangular waves as the control, supplemented the description of **Fig. 3f bottom panel** and the discussion about time-correlation between bionic spikes and bioelectrical signals;
6. Page 10-11, we have added the discussion about charge balancing and the detailed components of neuromorphic electrostimulator, supplemented the way to remove memory effect, clarified the concept on real-time monitoring of nerve stimulation, and emphasized the safety of electrode implantation;
7. Page 13, we have added the level of inflammatory factor in blood, counted the area of mitochondria and more comprehensive assessment of ES damage;
8. Page 14-15, we have performed comprehensive analysis on the effect of 2D ES: more inflammatory cytokines (TNF- α , IL-10 and IL-1 β) and additional control group (0.175 μ A t-ES) were added;
9. Page 16, we have supplemented and the definition of bionic spikes in **Table S1**;
10. Page 17, we have added a reference related to the role of NE to ADRB2, and added relevant discussions;

11. Page 18, we have repeated our experiment on transgenic mice with an antagonist of ADRB2 and added the discussion of *Lyz2-Cre* mice;
12. Page 21, we have added the definition of the SHAM group;
13. Page 22-23, we have supplemented the detail of anaesthesia during animal surgery, the tissue preparation for ELISA test, catalog number of ELISA kits for added inflammatory factors and the information of an antagonist of ADRB2;
14. Page 24-25, we have added the information of Cleaved Caspase-3 antibody, the detail of Calcein/PI staining;
15. Page 28-30, we have revised **Ref 18**, and added 6 references as needed;
16. Page 33, we have supplemented the blue and red curves of **Fig. 3f** with time axes;
17. Page 34, we have revised the **Fig. 4c** and **Fig. 4e right panel** and added the captions of tissue origin and the discussions of additional control group (0.175 mA t-ES).

Based on the reviewers' comments, we have performed the following revisions in the revised Supplementary Materials:

1. Page 2, we have added 5 new sections to the content and renumbered them;
2. Page 3, we have added a schematic of neuromorphic electrostimulator system;
3. Page 8, we have reorganized the figure of pulse widths and made it more visible;
4. Page 9, we have supplemented the finite element simulation of FGM device to fully show the working mechanism;
5. Page 13, we have emphasized that bionic stimulus spikes was collected under gate base voltage variation;
6. Page 14-15, we have supplemented the discussions of charge balancing in neuromorphic stimulation system;
7. Page 20, we have added the result of IL-6 in blood before and after implantation;
8. Page 21, we have made the scar bar more visible and added the mitochondrion area as an additional indicator of damage;
9. Page 22-24, we have supplemented comprehensive assessment of damage after ES including CC3 staining and Calcein/PI test;
10. Page 25, we have added stimulation threshold range of t-ES;
11. Page 27, we have comprehensively analyzed the inflammatory cytokines including TNF- α , IL-10, IL-1 β ;
12. Page 28, we have clarified that laser speckle was performed before, during and after 2D ES;
13. Page 31, we have provided the level change of IL-6 under the intervene of adrenoceptor;
14. Page 33, we have revised Table S1, including replacing the "hook electrodes" with "microcuff electrodes", "Specificity" with "Direct nerve stimulation" and "Real-time monitoring" with "real-time monitoring of nerve stimulation", and removing the reference for piezoelectric scaffolds by ultrasounds.

REVIEWER COMMENTS

Reviewer #1 (Remarks to the Author):

The authors have carefully addressed comments raised by reviewers, consolidating a new way for nerve stimulation. Some additional issues:

- 1) The involvement of surgical procedures to wrap this flexible 2D ES device around sympathetic ganglia, located in deep body tissue, is still very invasive, even through this new route, making this current stimulation method unlikely practical.

- 2) For SF24: IL-10 is supposed to be anti-inflammatory; please comment on the impact of the reduction of IL-10 by this 2D ES on inflammation control.

- 3) Current electroacupuncture (and traditional manual acupuncture) is not to put the electrodes directly onto the nerve, and the damage on nerves by those traditional ES may not as severe as the situation when the t-ES was applied directly onto the ganglia or cells (described in this study). The authors need to point out this and tune down the potential damage caused by t-ES, at least in terms of neuronal damage.

Reviewer #2 (Remarks to the Author):

Professor Zhou and his co-authors have diligently addressed all of the issues I raised about their manuscript entitled "Neuromorphic Electro-Stimulation Based on Atomically Thin Semiconductor for Damage-Free Inflammation Inhibition". They have provided comprehensive responses, revised the manuscript accordingly, and have improved the overall quality of the paper.

Response to Reviewers' Comments

Text in blue: Comments/questions from reviewers

Text in black: Response from the authors

Text in green: Text in the original submission

Text in red: Revisions made to the manuscript

Reviewer #1

General comments:

The authors have carefully addressed comments raised by reviewers, consolidating a new way for nerve stimulation. Some additional issues.

Response:

We are delighted with the reviewer's approval of our first round of response and greatly appreciate the reviewer's valuable comments and suggestions. Regarding the additional issues of implantation procedures and practical application mentioned by reviewer, we have described the **potential of bionic stimulation and the advantages of new implantation route to reduce invasiveness**, clarified that the proposed method is **still in the early stages of practical clinical application**, and further discussed **promising directions for clinical application**; As for anti-inflammatory cytokine IL-10 that also declined after 2D ES, we have **supplemented related discussions with possible reason**; In addition, as suggested by the reviewer, we have pointed out that **electro-acupuncture would not cause severe neuronal damage** compared with t-ES (also described other therapeutic applications of electro-acupuncture), and discussed **potential ways to tune down the damage associated with t-ES**.

We have responded to the reviewer's comments point by point and made revisions in the main text, which have been highlighted in yellow. **A summary of the revisions in response to the reviewer's comments can be found at the end of rebuttal letter.** We hope that the revised manuscript could address your additional concerns.

For detailed comments, the main responses are summarized as follows:

- The potential of bionic stimulation and the advantages of new implantation route to reduce invasiveness are described, and it is admitted that the current method is still in the early stages of practical clinical application, and promising directions for practical application are also discussed, which had been added to the main text;
- The possible reason for the decrease of anti-inflammatory cytokine IL-10 after 2D ES are explained and discussed;
- Pointed out that electro-acupuncture would not cause as severe neuronal damage compared to t-ES, described other treatment applications of electro-acupuncture, and discussed potential ways to tune down the damage associated with t-ES.

Comment 1:

The involvement of surgical procedures to wrap this flexible 2D ES device around sympathetic ganglia, located in deep body tissue, is still very invasive, even though this new route, making this current stimulation method unlikely practical.

Response:

We thank the reviewer for the valuable comment to help us further improve and refine the manuscript. On the one hand, we have described the potential of the proposed bionic stimulation and the advantages offered by the new implantation pathway for reduced invasiveness; at the same time, we agree with the reviewer that the current method is still some distance from practical clinical application and discuss possible future directions for practical application. **In order to more accurately describe the fact that the proposed stimulation method requires an invasive procedure and is still in the early stages for practical clinical application, we have added relevant discussions in the revised manuscript.**

In this work, we have proposed a potential ES solution based on atomically thin MoS₂ floating-gate memory interdigital circuits and flexible electrodes, which allows

for direct programmable bionic stimulation to the target nerves, thereby inhibiting inflammation with lower currents. Although the implantation procedure required to wrap the nerve in deep body tissue for direct stimulation is still invasive, as mentioned by the reviewer, we have **used a new implantation pathway compared to traditional abdominal implantation to reduce the invasiveness** (hemorrhage, inflammation, etc.) as much as possible. One fact is that **the implanted mice could survive for at least 4 weeks**, and tests between SHAM (sham surgery only without electrode implantation) and sham (sham surgery and electrode implantation without ES) groups showed that **the implantation did not pre-introduce inflammation (Figure R1) to interfere with subsequent animal models.**

The proposed ES solution demonstrates the potential of bionic stimulation, but due to limitations such as scaled integration on the device side and invasiveness on the biological surgery side, **we agree with the reviewer that it is still in its infancy and a long way from practical clinical application.** Neurostimulation by minimally invasive surgery of wirelessly powered flexible chips with integrated floating-gate memory circuits and electrodes will be our future endeavor to make it possible for clinical application. In addition, non-invasive transcutaneous bionic stimulation may also be a promising application scenario for the treatment of symptoms such as persistent hiccups, systemic lupus erythematosus, and atrial fibrillation (**Ref 1-3**), which will also be of interest to us in the future.

(Supplementary Figure 17) Figure R1. The level of IL-6 in blood before and after implantation. The result between SHAM and sham groups showed that implantation had a negligible effect on the elevation of inflammatory cytokines. ns, no significance.

References:

1. Schulz-Stübner S, Kehl F. Treatment of persistent hiccups with transcutaneous phrenic and vagal nerve stimulation[J]. Intensive care medicine, 2011, 37: 1048-1049.
2. Aranow C, Atish-Fregoso Y, Lesser M, et al. Transcutaneous auricular vagus nerve stimulation reduces pain and fatigue in patients with systemic lupus erythematosus: a randomised, double-blind, sham-controlled pilot trial[J]. Annals of the rheumatic diseases, 2021, 80(2): 203-208.
3. Stavrakis S, Humphrey M B, Scherlag B J, et al. Low-level transcutaneous electrical vagus nerve stimulation suppresses atrial fibrillation[J]. Journal of the American College of Cardiology, 2015, 65(9): 867-875.

Related discussions have been added to the revised manuscript:

(Page 20 line 6-12) However, it is a fact that the proposed bionic stimulation solution is still in its infancy for practical clinical applications due to the limitations of scale integration in terms of devices and invasiveness in terms of biological surgery. And implantation of wirelessly powered flexible chips with integrated floating-gate memory circuits and electrodes via minimally invasive surgery, as well as non-invasive

transcutaneous stimulation will be possible directions for neuromorphic bionic ES solution towards practical clinical applications.

Comment 2:

For SF24: IL-10 is supposed to be anti-inflammatory; please comment on the impact of the reduction of IL-10 by this 2D ES on inflammation control.

Response:

We thank the reviewer for this comment. We suggest that **the reduction in IL-6 and IL-10 show the effectiveness of 2D ES in inhibiting the progress of inflammation**, and **the reduction in the anti-inflammatory cytokine IL-10 may be attributable to the decrease in the inflammatory cytokine IL-6**, as IL-6 could promote the secretion of IL-10. In the case of reduced IL-10, we have added relevant discussions in the revised manuscript.

As mentioned by the reviewer, **IL-10 is an anti-inflammatory cytokine** that increases when inflammation is present. For example, human Kupffer cells secrete IL-10 in response to a lipopolysaccharide challenge (**Ref 1**). In addition, the inflammatory cytokine IL-6 has been reported to promote IL-10 production by T cells, which also reflects the presence of inflammation (**Ref 2**). In this work, both IL-6 and IL-10 decreased after 2D ES, which **suggests that 2D ES is effective in preventing the progress of inflammation**. Since a **significant decline** in the inflammatory cytokine IL-6 would occur with suppressed inflammation, while the anti-inflammatory cytokine IL-10 had **limited elevation**. And the decrease in IL-10 may be attributed to the significant decrease in IL-6, which has been shown to facilitate IL-10 secretion according to **Ref 2**.

Refernece:

1. Knoll P, Schlaak J, Uhrig A, et al. Human Kupffer cells secrete IL-10 in response to lipopolysaccharide (LPS) challenge[J]. Journal of hepatology, 1995, 22(2): 226-229.

2. McGeachy M J, Bak-Jensen K S, Chen Y I, et al. TGF- β and IL-6 drive the production of IL-17 and IL-10 by T cells and restrain TH-17 cell-mediated pathology[J]. Nature immunology, 2007, 8(12): 1390-1397.

Related discussions have been added to the revised manuscript:

(Page 15 line 7-12) Although acting as an anti-inflammatory cytokine, IL-10 also appeared to decrease after 2D ES, because in the case of suppressed inflammation, there is a significant reduction in inflammatory cytokine IL-6, and the elevation of anti-inflammatory cytokine IL-10 is limited. And IL-6 could promote the secretion of IL-10⁴⁸, the significantly reduced IL-6 may have led to the reduction in IL-10, which further suggests the effectiveness of 2D ES in inhibiting the progress of inflammation.

Comment 3:

Current electroacupuncture (and traditional manual acupuncture) is not to put the electrodes directly onto the nerve, and the damage on nerves by those traditional ES may not as severe as the situation when the t-ES was applied directly onto the ganglia or cells (described in this study). The authors need to point out this and tune down the potential damage caused by t-ES, at least in terms of neuronal damage.

Response:

We thank the reviewer for the kind and constructive comments. In the revised manuscript, we have **pointed out that electro-acupuncture would not cause as severe neuronal damage compared to t-ES**, described other treatment applications of electro-acupuncture, and **discussed potential ways to tune down the damage associated with t-ES**.

We agree with the reviewer that **traditional ES based on electro-acupuncture or manual acupuncture does not cause as much damage to nerves or neurons as t-ES**. Because electro-acupuncture or manual acupuncture is usually performed in somatic tissue, with stimulation needles placed far away from the target nerve and not directly onto the nerve bundles, while the t-ES involved in this work uses a commercial

stimulator to deliver high current stimulation directly to the nerve (sympathetic ganglia) through flexible electrodes. **We have revised the relevant discussions in the manuscript and pointed this out.** Furthermore, electro-acupuncture mainly acts on acupoints, and different acupoints correspond to different therapeutic effects. In addition to anti-inflammation, **electro-acupuncture-based ES also shows great potential for treating other diseases.** For example, stimulation of bilateral BL33 (Zhongliao) and BL35 (Huiyang) improves urinary leakage (**Ref 1**); ST36 (Zusanli), ST37 (Shangjuxu) and ST39 (Xiajuxu) treats postoperative ileus (**Ref 2**); and CV23 (Lianquan) acupoint has been shown to be beneficial for dysphagia (**Ref 3**).

As for the potential damage from t-ES, which is closely related to the amplitude and duration of the stimulation current. **Lower stimulation currents or shorter stimulation durations are expected to reduce neuronal damage caused by t-ES, while stimulation efficacy may also be degraded (Figure R2).** For t-ES, it is difficult to find a trade-off between low damage and effectiveness. And the proposed 2D ES becomes an alternative solution that uses neuromorphic bionic stimulation to achieve efficient inflammation inhibition at lower stimulation currents than t-ES (**Figure R2**), while not causing neuronal damage.

(Figure 4c) Figure R2. ELISA for IL-6 in the non-ES, t-ES and 2D ES groups. The level of IL-6 was decreased by 73.5% in the 2D ES group compared with the non-ES group. the 2D ES group with average currents of ~0.175 mA resulted in a further 70.6% drop in IL-6 levels compared to the t-ES group with fixed currents at 0.175 mA, which suggests that the specific neuromorphic bionic spikes lowered the threshold for ES efficacy. Moreover, the level of IL-6 had no significant statistical difference between t-ES of 0.3 mA and 2D ES of 0.175 mA, which means that 2D ES achieved the same effect of t-ES but with a 41.8% lower stimulation current. n=5-6. ns, no significance; *, $p < 0.05$; **, $p < 0.01$; ****, $p < 0.0001$.

Refernece:

1. Liu Z, Liu Y, Xu H, et al. Effect of electroacupuncture on urinary leakage among women with stress urinary incontinence: a randomized clinical trial[J]. *Jama*, 2017, 317(24): 2493-2501.
2. Chen K B, Huang Y, ** X L, et al. Electroacupuncture or transcutaneous electroacupuncture for postoperative ileus after abdominal surgery: a systematic review and meta-analysis[J]. *International Journal of Surgery*, 2019, 70: 93-101.
3. Yao L, Ye Q, Liu Y, et al. Electroacupuncture improves swallowing function in a post-stroke dysphagia mouse model by activating the motor cortex inputs to the nucleus tractus solitarii through the parabrachial nuclei[J]. *Nature Communications*, 2023, 14(1): 810.

Related discussions have been added to the revised manuscript:

(Page 3 line 19-Page 4 line 2) Currently, electro-acupuncture or manual acupuncture is commonly used for stimulation, and although it is minimally invasive, this non-directional stimulation requires higher current amplitudes. However, electro-acupuncture stimulation would not cause severe damage to neurons, as stimulation needles are usually placed in somatic tissue, away from the nerve bundle. While the lack of direct nerve stimulation reduces efficacy and leads to the demand for higher stimulation currents.

(Page 14 line 1-6) The above results of HE and TEM showed that although 0.3 mA reduced inflammatory cytokines, it also introduces damage to SChG, which can lead to complications (It is worth pointing out that electro-acupuncture would not cause neuronal damage as severe as t-ES, since the stimulation needles are usually placed in somatic tissue and does not act directly on the nerve bundles). Lower currents or shorter stimulation durations are expected to tune down the damage caused by t-ES, but the efficacy may also be degraded.

Reviewer #2

General comments:

Professor Zhou and his co-authors have diligently addressed all of the issues I raised about their manuscript entitled "Neuromorphic Electro-Stimulation Based on Atomically Thin Semiconductor for Damage-Free Inflammation Inhibition". They have provided comprehensive responses, revised the manuscript accordingly, and have improved the overall quality of the paper.

Response:

We are pleased that the reviewer was satisfied with our response and thank the reviewer again for the valuable suggestions and efforts to improve this work.

Based on the reviewers' comments, we have performed the following revisions in the revised main text:

1. Page 3-4, we have revised the description of electro-acupuncture to state that it would not cause severe neuronal damage;
2. Page 14, we further point out that electro-acupuncture would not cause severe damage to neurons compared with t-ES and discuss potential ways to tune down the damage accompanied by t-ES;
3. Page 15, we explain and discuss the possible reason why the anti-inflammatory cytokine IL-10 also showed a decrease after 2D ES;
4. Page 20, we clarify that the proposed stimulation solution is still in the early stages of practical clinical application and further discuss promising directions for future clinical application;
5. Page 31, We added a reference (Ref 48) about the inflammatory cytokine IL-6 promoting the secretion of the anti-inflammatory cytokine IL-10.

REVIEWERS' COMMENTS

Reviewer #1 (Remarks to the Author):

The authors have carefully addressed reviewers' comments